# Discovering Generalizable Governing Equations for Graph Dynamical Systems with Interpretable Neural Networks

**Riccardo Cappi**                                                          *riccardo.cappi@phd.unipd.it*
*University of Padua, Padua, Italy*

**Paolo Frazzetto**                                                          *paolo.frazzetto@unipd.it*
*University of Padua, Padua, Italy*

**Nicolò Navarin**                                                          *nicolo.navarin@unipd.it*
*University of Padua, Padua, Italy*

**Alessandro Sperduti**                                                          *alessandro.sperduti@unipd.it*
*University of Padua, Padua, Italy*
*Fondazione Bruno Kessler, Trento, Italy*

**Reviewed on OpenReview:** *https://openreview.net/forum?id=a2mPNSSAYL*

## Abstract

The discovery of symbolic governing equations is a central goal in science; yet, it remains challenging particularly for graph dynamical systems, where the network topology further shapes the system behavior. While artificial intelligence offers powerful tools for modeling these dynamics, the field lacks a rigorous comparative benchmark to assess the true scientific utility of the discovered laws. To address this challenge, this work proposes a novel evaluation pipeline designed to rigorously assess state-of-the-art symbolic regression models for graph equation discovery. Moving beyond simple fitting metrics, this framework evaluates discovered laws based on their long-term trajectory stability and, critically, their out-of-distribution generalization to unseen graph topologies. We benchmark established methods, including sparse regression and MLP-based architectures, and introduce the Graph Kolmogorov-Arnold Network-ODE (GKAN-ODE) model, a novel adaptation of KANs explicitly tailored for this domain, augmented by hyperparameter-free multiplicative nodes and a new Spline-Wise symbolic regression algorithm. Across a suite of synthetic and real-world graph dynamical systems, we numerically demonstrate through extensive experiments that neural-based approaches, particularly the GKAN-ODE model, recover exact ground-truth equations and achieve trajectory errors up to two orders of magnitude lower than the baseline methods on out-of-distribution test graphs.

## 1 Introduction

The pursuit of scientific knowledge is undergoing a profound transformation, driven by the confluence of vast datasets and sophisticated computational tools. In this "Fourth Paradigm" of science (Hey et al., 2009), Artificial Intelligence (AI) promises not only to accelerate discovery but also to fundamentally change its nature (Wang et al., 2023). The vision extends beyond creating models with high predictive accuracy; the true frontier lies in developing AI that can help us understand the world, unveiling the underlying principles and causal mechanisms that govern complex phenomena (Camps-Valls et al., 2023). This ambition, however, is often hindered by the "black-box" nature of deep learning models, whose internal workings are largely opaque, creating a barrier between computational power and human understanding (Rudin, 2019).

This is challenging, especially in the study of *graph dynamical systems* (Barrat et al., 2008; Liu et al., 2025a). These systems, where entities interact with each other and evolve according to local interactions on a network,

are ubiquitous in science, from gene regulatory networks and neural circuits, to the spread of epidemics and social dynamics (Barabási, 2013). While we can often observe their temporal evolution, the fundamental laws governing their behavior often remain unknown and are heavily dependent on the specific graph instance. Our central objective is to move beyond mere simulation by discovering the symbolic governing *Ordinary Differential Equations* (ODEs) that dictate the evolution of node states directly from observational data.

*Symbolic Regression* (SR) (Makke & Chawla, 2024) emerges as the natural instrument for this task. While traditional evolutionary algorithms and modern sparsity-based frameworks have laid crucial groundwork, the advent of deep learning has opened new possibilities. *Neural Networks* (NNs), with their ability to approximate arbitrary nonlinear functions, can learn the underlying dynamics with high fidelity. However, this expressivity typically comes at the cost of interpretability, requiring a separate post-hoc SR step to distill symbolic knowledge from the opaque models (Cranmer, 2023).

Despite these advances, a critical gap persists in the literature. The landscape of neural-based equation discovery for graph dynamics is fragmented, with only a handful of dedicated methods (Gao & Yan, 2022; Hu et al., 2025) and with no systematic comparative assessment of their performance under different conditions. Existing evaluations typically validate the discovered laws on trajectories generated from the same graph instance used for training, without considering the generalization capabilities of the discovered models to graph topologies unseen during training. Furthermore, the post-hoc SR step applied to neural-based models is often performed without systematic hyperparameter validation, so a reported equation may reflect a fortunate hyperparameter choice rather than a reproducible procedure. Researchers seeking to apply these powerful tools, therefore, lack a clear reference for which architecture to choose, how to implement it, and how to evaluate the scientific plausibility of the discovered equations. Finally, the potential of a novel and interpretable-by-design architecture like *Kolmogorov-Arnold Networks* (KANs) (Liu et al., 2025b) remains unexplored in this field, despite their demonstrated potential for scientific discovery in other domains (Liu et al., 2024; Koenig et al., 2024). This paper aims to fill this gap. We present a rigorous, comparative study designed to unveil the actual performance of neural-based models for equation discovery on graph dynamical systems. Our contributions are fourfold:

1. **We provide a rigorous and reproducible evaluation pipeline** of state-of-the-art methods, including a leading sparse regression algorithm and *Multilayer perceptron-based* architectures (MLPs), for assessing equation discovery on graphs. By making our code and experimental setup publicly available, we establish a firm baseline for future research [1].

2. **We introduce the Graph KAN-ODE (GKAN-ODE)**, a novel adaptation of Kolmogorov-Arnold Networks for graph dynamics. We enhance the standard architecture with hyperparameter-free multiplicative nodes to better capture physical interactions and propose a principled, structure-aware *Spline-Wise* symbolic regression algorithm to distill faithful formulas directly from KAN architectures.

3. **We conduct extensive experiments** on both synthetic systems with known ground truths and challenging real-world epidemic data. Our evaluation hinges on a stringent **long-term trajectory rollout metric**, which assesses the stability of the discovered laws that go beyond simple one-step prediction accuracy. Moreover, we demonstrate that the learned symbolic models **generalize effectively to out-of-distribution settings** in unseen scenarios, highlighting their robustness and scientific plausibility. We show that GKAN-ODE recovers exact ground-truth equations, outperforming existing baselines by up to two orders of magnitude in trajectory error, while using fewer parameters.

4. **We offer a critical analysis of the expressivity-interpretability trade-off**. By comparing black-box and white-box symbolic distillation strategies, we provide practical observations for researchers, clarifying how model choice impacts the complexity and scientific plausibility of the discovered laws.

---

[1]The code is available at https://github.com/riccardocappi/Kan-for-Interpretable-Graph-Dynamics

This work, therefore, serves as both a methodological contribution and a comprehensive guide, aimed at facilitating the discovery of governing laws from observational data on complex networked systems.

## 2 Related Works

### 2.1 Symbolic Regression for Scientific Discovery

Symbolic regression is a methodology for discovering closed-form mathematical expressions from data. Let $\mathcal{O}$ denote a finite set of admissible primitive operators (e.g., $\mathcal{O} = \{+, -, \times, \div, \sin, \exp, \log, \ldots\}$). We denote by $\mathcal{F}_{\mathcal{O}}$ the hypothesis space of all closed-form expressions constructible via finite composition of elements in $\mathcal{O}$ and real-valued constants. While parametric regression restricts the search to parameter estimations $\theta \in \mathbb{R}^p$ for a fixed functional form $g(\,\cdot\,; \theta)$, SR traverses the combinatorial space $\mathcal{F}_{\mathcal{O}}$ to identify the functional form itself. Formally, given a dataset of input-output pairs $\mathcal{D} = \{(\mathbf{x}_i, y_i)\}_{i=1}^M$ generated by an unknown target function $f : \mathbf{x} \mapsto y$, an SR algorithm defines a mapping:

$$\text{SR} : \mathcal{D} \longmapsto \hat{f}_{\text{SR}} \in \mathcal{F}_{\mathcal{O}}, \quad \text{such that} \quad \hat{f}_{\text{SR}}(\mathbf{x}) \approx f(\mathbf{x}) = y \; \forall \, \mathbf{x} \in \mathcal{D}, \tag{1}$$

subject to a structural complexity constraint on $\hat{f}_{\text{SR}}$.

Historically, this field was dominated by evolutionary methods like *Genetic Programming* (GP) (Schmidt & Lipson, 2009; Cranmer, 2023), which, while powerful, often face scalability challenges. A prominent alternative is the *Sparse Identification of Nonlinear Dynamics* (SINDy) framework (Brunton et al., 2016), which recasts equation discovery as a sparse regression problem over a library of candidate functions. For network systems, TPSINDy extends this by modeling the system's dynamics as a two-part sparse regression problem, finding separate expressions for the self-dynamics and interaction components (Gao & Yan, 2022).

### 2.2 Deep Learning for Equation Discovery on Graphs

One of the first attempts to leverage NNs to learn analytical expressions was the development of equation learner (EQL) networks (Martius & Lampert, 2017), in which non-linear activation functions are replaced by primitive functions, analogous to SR. Another remarkable work is *AI-Feynman* (Udrescu et al., 2020), an algorithm that combines SR and NN fitting with a suite of physics-inspired techniques that outperformed previous benchmarks. A pivotal contribution by Cranmer et al. (2020) showed that *Graph Neural Networks* (GNNs) can effectively learn the dynamics of systems of particles, and their learned latent representations can then be distilled into symbolic expressions via post-hoc SR. The recent *Learning Law of Changes* (LLC) framework (Hu et al., 2025) advances this approach for graph dynamical systems. It employs separate MLPs to model the self-dynamics and interaction terms (with an explicit multiplicative bias) and then parses them into symbolic form using a post-hoc SR step. Their results demonstrate significant performance gains over prior SR techniques for network dynamics, establishing a key state-of-the-art contribution. Graph neural-ODE models (Poli et al., 2019) sit at one extreme of this diagram. They are highly expressive, as they represent the learning task (not per se a graph dynamical system) in a high-dimensional latent space that evolves over time, following the Neural-ODE approach. However, these internal representations are opaque and yield no symbolic insight. At the opposite end lies TPSINDy (Gao & Yan, 2022; Navarin et al., 2024), which is interpretable by construction but is constrained to the functional forms in its predefined library of symbolic terms. The neural-based approaches (GKAN-ODE and LLC) attempt to balance these two extremes by training a flexible neural model and then performing an SR step over its input-output behavior to distill a mathematical expression of the learned function. The proposed GKAN-ODE moves further along the interpretability axis compared to LLC, since the KAN architecture exposes each univariate activation function individually (more details in Section 2.3), allowing direct inspection of the model's internal components and enabling structure-aware symbolic regression, which is not meaningful with MLP-based architectures. Figure 1 shows different methods for learning on graph dynamical systems in a qualitative manner (assuming comparably trained models with an identical compute budget) in terms of expressivity and interpretability, mirroring the style proposed by Koenig et al. (2024).

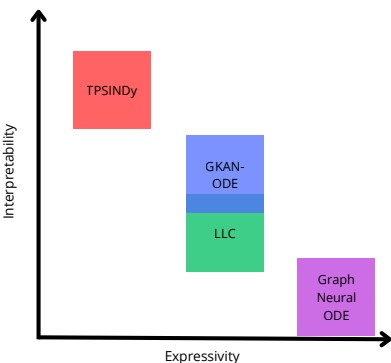

Figure 1: Qualitative diagram comparing existing methods for learning on graph dynamical systems.

## 2.3 Kolmogorov-Arnold Networks: A Path Towards Interpretability

Kolmogorov-Arnold Networks (KANs), proposed by Liu et al. (2025b), are a specific type of neural network that has been recently proposed as a valid alternative to Multi-Layer Perceptrons (MLPs). Whereas MLPs are inspired by the universal approximation theorem, KANs are inspired by the Kolmogorov-Arnold representation theorem (Kolmogorov, 1961; Braun & Griebel, 2009), which states that, given a smooth function $f : [0,1]^d \rightarrow \mathbb{R}$, it can be written as:

$$f(\mathbf{x}) = f(x_1, x_2, ..., x_d) = \sum_{q=1}^{2d+1} \Phi_q \left( \sum_{p=1}^{d} \phi_{q,p}(x_p) \right), \tag{2}$$

where $\phi_{q,p} : [0,1] \rightarrow \mathbb{R}$, $\Phi_q : \mathbb{R} \rightarrow \mathbb{R}$. In other words, $f$ can be reduced to a suitably defined composition of univariate functions, where the composition only involves simple addition. The underlying idea of KANs is to substitute the weights and fixed activation functions of MLPs with learnable univariate activation functions on edges and sum aggregation on nodes.

The general definition of a KAN layer $\mathbf{\Phi}_l$ with $d_{in}$-dimensional input and $d_{out}$-dimensional output consists of a matrix of univariate functions:

$$\mathbf{\Phi}_l = \begin{bmatrix} \phi_{l,1,1}(\cdot) & \phi_{l,1,2}(\cdot) & \cdots & \phi_{l,1,d_{in}}(\cdot) \\ \phi_{l,2,1}(\cdot) & \phi_{l,2,2}(\cdot) & \cdots & \phi_{l,2,d_{in}}(\cdot) \\ \vdots & \vdots & \ddots & \vdots \\ \phi_{l,d_{out},1}(\cdot) & \phi_{l,d_{out},2}(\cdot) & \cdots & \phi_{l,d_{out},d_{in}}(\cdot) \end{bmatrix} \tag{3}$$

where $\phi_{l,j,i}$ represents the learnable activation function applied to the $i^{th}$-feature of the input of the $j^{th}$-neuron at layer $l$. After computing all the $d_{in} \cdot d_{out}$ activation values, the output of the $l^{th}$ layer $\mathbf{x}_l \in \mathbb{R}^{d_{out}}$ is obtained by summing along the first dimension of the matrix described in Equation 3. Stacking multiple KAN layers results in an architecture with a shape represented by an integer array $[d_0, d_1, ..., d_L]$, where $d_l$ represents the number of neurons in the $l^{th}$-layer and $d_0 = \dim(\mathbf{x})$, i.e., the dimensionality of $\mathbf{x}$. Each univariate function in Equation 3 is defined as a B-spline with residual activation and trainable control points that can be learned through backpropagation and gradient descent. Other possible parametrizations for KANs' activations include radial basis functions (Chao et al., 2026; Li, 2024), Chebyshev polynomials (SS et al., 2024), or wavelet functions (Seydi et al., 2024).

This design shifts the complexity from matrix multiplications and nonlinear activations to a set of univariate functions that can be individually visualized, analyzed, and symbolically regressed to provide a human-readable mathematical formulation of the function learned by the model. Further technical details can be found in the original paper or in Appendix A.1. The potential of KANs for scientific discovery has been demonstrated in learning PDE solutions (Liu et al., 2024) and discovering physical laws in dynamical systems without an explicit interaction structure (Koenig et al., 2024). KANs are particularly well-suited for this

domain for mainly two reasons: (i) Physical governing equations are generally composed of smooth, structured nonlinear terms, such as trigonometric functions in oscillator models, exponentials in reaction kinetics, or power laws in population dynamics. A network whose elementary building blocks are themselves smooth, learnable nonlinear functions is therefore naturally aligned with this class of targets. (ii) Once training is complete, each univariate activation can be inspected and matched to a symbolic function, providing a transparent, node-by-node view of the model's learned dynamics. However, to our knowledge, KANs have not yet been applied to discover the governing equations of graph dynamical systems, where network topology drives the evolution of node states over time. Their use has been limited to other graph-based tasks (Bresson et al., 2025), not to the specific challenge of discovering underlying temporal dynamics.

## 3 Methods

This section details our proposed evaluation pipeline for equation discovery. We first establish the formal context for our work by defining graph dynamical systems. Next, we describe the general neural training pipeline, then introduce our Graph KAN-ODE (GKAN-ODE) architecture, and finally outline the symbolic regression procedures and evaluation protocol.

### 3.1 Mathematical Formulation and Notation

The systems under investigation are graph dynamical systems or dynamical processes on complex networks. Such a system is defined by a graph $\mathcal{G} = (\mathcal{V}, \mathcal{E})$, where $\mathcal{V}$ is a set of $N$ nodes (or components) and $\mathcal{E}$ is a set of edges representing their interactions. The state of each node $i \in \mathcal{V}$ at time $t \in \{0, \dots, T\}$ is described by a vector $\mathbf{x}_i(t) \in \mathbb{R}^d$, while the whole system state is defined as $\mathbf{X}(t) \in \mathbb{R}^{N \times d}$. The graph topological structure can be represented by the adjacency matrix $A \in \mathbb{R}^{N \times N}$, where each entry denotes the connection strength between nodes $i$ and $j$, and $A_{ij} = 0 \iff e_{ij} \notin \mathcal{E}$. As in related works, we focus on graphs with *static* topology, where $\forall\, t,\; A(t) = A$, and in a time-invariant context in which the temporal dynamics of a node $\mathbf{x}_i(t)$ is described by an autonomous ODE:

$$\frac{d\mathbf{x}_i}{dt} = f\left(\mathbf{x}_i, \{\mathbf{x}_j\}_{j \in \mathcal{N}(i)}\right) = \dot{\mathbf{x}}_i \quad \forall\, t, \tag{4}$$

where $\mathcal{N}(i)$ denotes the neighborhood of node $i$. For clarity, we will omit the explicit time dependence of $\mathbf{x}_i(t)$ hereafter, unless when denoting data points. Following the principle of universality in network dynamics (Barzel & Barabási, 2013) for pairwise interactions, the governing function $f$ can be decomposed into two fundamental components: an intrinsic *self-dynamics* function $H : \mathbb{R}^d \to \mathbb{R}^d$ and an *interaction* function $G : \mathbb{R}^d \times \mathbb{R}^d \to \mathbb{R}^d$ that aggregates effects from neighboring nodes. The dynamics of any node $i$ can thus be expressed as:

$$\dot{\mathbf{x}}_i = H(\mathbf{x}_i) + \sum_{j=1}^{N} A_{ij}\, G(\mathbf{x}_i, \mathbf{x}_j). \tag{5}$$

The primary objective of this work is to discover the symbolic forms of both $H$ and $G$ from discrete-time observations $\{\mathbf{X}(t)\}_{t=0}^{T}$. Models and estimated quantities are denoted with a hat, e.g., $\hat{H}, \hat{\mathbf{x}}_i$.

### 3.2 Learning Dynamics on Graphs with Neural Models

Our primary data consist of time series of graph states $\{\mathbf{X}(t)\}_{t=0}^{T}$, representing discrete measurements of an underlying continuous process. As a prerequisite for learning, we require an estimate of the instantaneous rate of change, the time derivative $\dot{\mathbf{X}}(t)$. We compute a numerical value of the time derivative for each node $\mathbf{x}_i$ using the five-point stencil method (Gao & Yan, 2022), a choice that balances accuracy with robustness to noise in the observational data. The five-point stencil is applied to interior time steps $t \in \{2, \dots, T-2\}$, while derivative estimates at the boundary steps $t \in \{0, 1, T-1, T\}$ are obtained via first and second-order forward and backward finite differences, respectively. This yields a corresponding sequence of derivative evaluations $\{\dot{\mathbf{X}}(t)\}_{t=0}^{T}$. We then train a neural network to learn the mapping from the system's state $\mathbf{X}(t)$ to its derivative $\dot{\mathbf{X}}(t)$. Specifically, following the decoupled formulation in Equation 5, we parameterize the self-dynamics $H$ and interaction dynamics $G$ with two distinct neural networks, $\hat{H}$ and $\hat{G}$, defined by sets

of learnable parameters $\theta_{\hat{H}}$ and $\theta_{\hat{G}}$, respectively. The models are trained via gradient descent to minimize the Mean Absolute Error (MAE) loss function between the numerically estimated derivatives $\dot{\mathbf{X}}(t)$ and the model's predictions $\hat{\dot{\mathbf{X}}}(t)$ over the entire training set:

$$\mathcal{L}_{train}(\theta_{\hat{H}}, \theta_{\hat{G}}) = \frac{1}{N(T+1)} \sum_{i=1}^{N} \sum_{t=0}^{T} |\dot{\mathbf{x}}_i(t) - \hat{\dot{\mathbf{x}}}_i(t)|. \tag{6}$$

The quantities $N$ and $T$ are dataset-dependent and are reported per experiment in Appendix B.1.

### 3.3 Graph Kolmogorov-Arnold Networks for ODE Discovery

We propose and assess a novel approach, the GKAN-ODE model, where functions $\hat{H}$ and $\hat{G}$ are parameterized by distinct KANs. We parametrize KANs' activations as B-splines, inheriting the original formulation proposed by Liu et al. (2025b). For convenience, throughout this work we use the term *splines* to denote the generic KAN activation functions. In line with the principle that physical laws are often sparse (Brunton et al., 2016), we include the KAN-specific $L^1$ sparsity penalty Liu et al. (2025b) (details can be found in Appendix A.1) to encourage both $\hat{H}$ and $\hat{G}$ networks to prune inactive splines.

To better capture the multiplicative relationships common in physical dynamics, we further enhance the standard KAN architecture. In fact, previous work has introduced dedicated multiplication layers *between* KAN layers (Liu et al., 2024), which however lead to the addition of structural hyperparameters, which require prior knowledge or extensive tuning. To circumvent this, we propose a more integrated extension where multiplication occurs *within* each KAN layer. Specifically, for a KAN layer with $d_{out}$ output neurons, we designate half $\lceil d_{out}/2 \rceil$ as standard additive nodes and the remaining $\lfloor d_{out}/2 \rfloor$ as multiplicative nodes, where the input univariate splines are multiplied rather than summed. An example of a KAN architecture with the proposed multiplicative enhancement is shown in Figure 2a. This design allows the model itself, guided by data and sparsity, to learn the appropriate functional form (additive, multiplicative, or a combination) without additional hyperparameters. The reason for the introduction of multiplicative nodes lies in the fact that, by default, a KAN is an additive model (Figure 2c), meaning that each neuron is defined as a sum of univariate functions. In this case, learning a multiplicative interaction $x_1 x_2$ requires the network to exploit indirect representations. For example, a [2, 1, 1] KAN can learn $x_1 x_2$ via a logarithmic composition (Figure 2d): the intermediate neuron computes $\ell_1 = \ln(x_1) + \ln(x_2)$, and the output activation learns $e^{\ell_1}$. Now consider the same [2, 1, 1] KAN, but with the intermediate neuron replaced by our proposed multiplicative node (Figure 2b). Instead of learning $\ln(x_1)$ and $\ln(x_2)$, the two input splines only need to learn identity functions, and the multiplicative node computes $\ell_1 = x_1 x_2$ directly. The output neuron's activation then needs to learn a simple linear function, making the learning task substantially simpler. Our empirical findings confirm that sparse training effectively prunes multiplicative nodes when the dynamics are purely additive and retains them when they are essential. In Appendix C.3, we further discuss the benefits of multiplicative nodes over the original architecture, especially in the context of symbolic distillation.

### 3.4 Symbolic Regression Procedures

Once a neural model is trained, we extract symbolic formulas using two distinct strategies: a model-agnostic, black-box approach and a structure-aware, white-box approach exclusive to KANs.

#### 3.4.1 Black-Box Symbolic Regression

A black-box SR method (Cranmer, 2023) takes data and a model as input and produces symbolic expressions approximating the model predictions. Notably, this procedure treats the models as opaque functions, making it applicable to any machine learning method. In our case, given the trained neural networks $\hat{H}$ and $\hat{G}$, we first generate input-output pairs by performing a forward pass over the training data: $\{\mathbf{x}_i(t), \hat{H}(\mathbf{x}_i(t))\}$ and $\{(\mathbf{x}_i(t), \mathbf{x}_j(t)), \hat{G}(\mathbf{x}_i(t), \mathbf{x}_j(t))\}$ for all interacting pairs. We then fit a separate SR model to each set to obtain symbolic expressions $\hat{H}_{SR}$ and $\hat{G}_{SR}$:

$$\text{SR}(\{\mathbf{x}, \hat{H}(\mathbf{x})\}) = \hat{H}_{SR}, \quad \text{SR}(\{(\mathbf{x}_i, \mathbf{x}_j), \hat{G}(\mathbf{x}_i, \mathbf{x}_j)\}) = \hat{G}_{SR}. \tag{7}$$

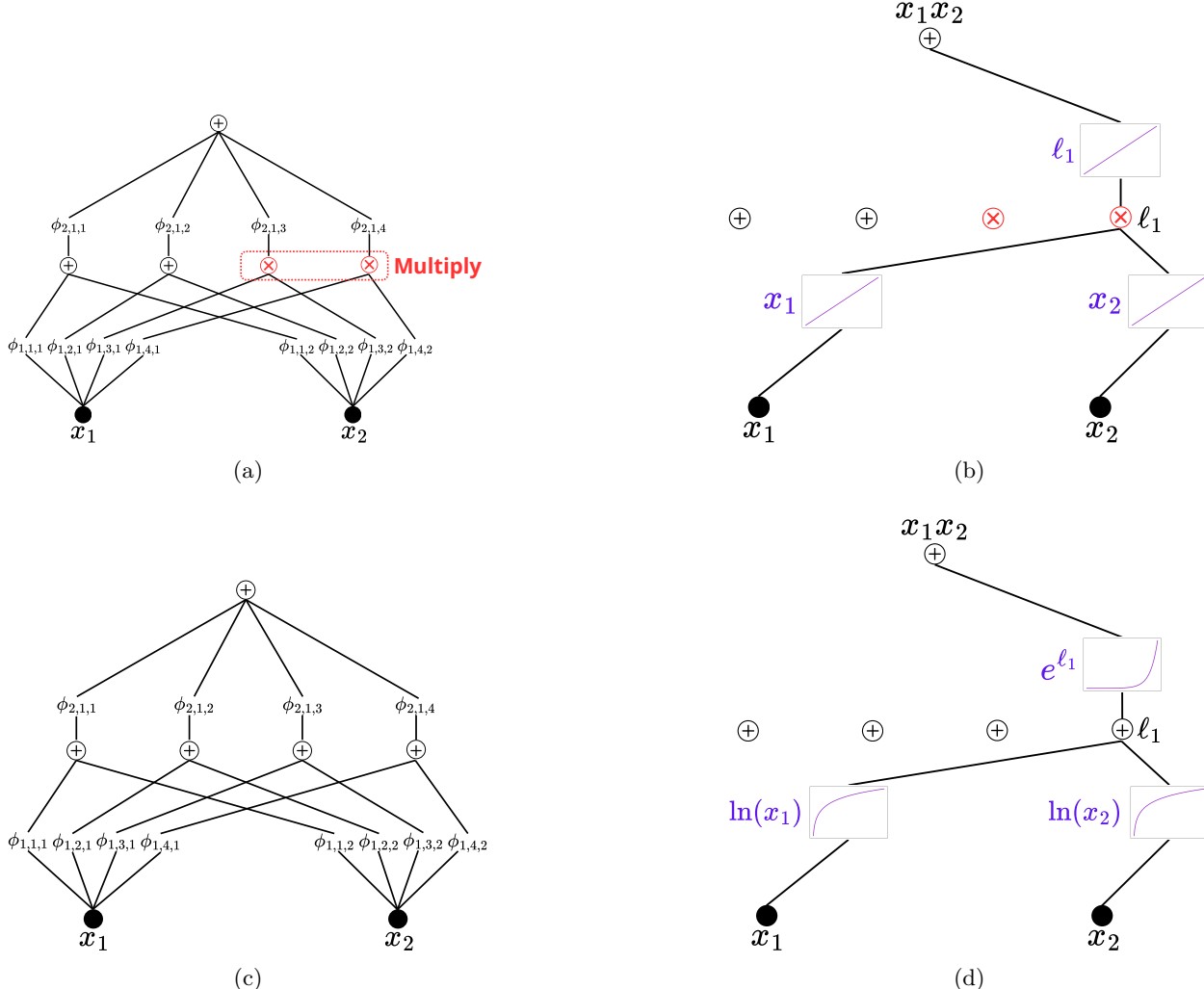

Figure 2: (a) Representation of a [2, 4, 1] KAN architecture with the proposed multiplication enhancement (red) for a two-dimensional input ($d = 2$). (b) Pruned [2, 1, 1] KAN learning the multiplicative term $x_1 x_2$ through the proposed multiplicative node. (c) Default KAN additive architecture for a two-dimensional input. (d) Pruned [2, 1, 1] KAN learning $x_1 x_2$ through logarithmic composition. $\phi$ are the univariate spline activations.

The final symbolic model of the full ODE, $f_{SR} \approx \dot{\mathbf{x}}_i$, is constructed by composing these two discovered expressions according to the governing structure of Equation 5.

### 3.4.2 Spline-Wise Symbolic Regression for KANs

The architecture of KANs enables a more granular and transparent approach: instead of regressing on the network's aggregate output, we can distill expressions from its elementary components, i.e., the univariate spline activations $\phi$. To fully leverage the transparent structure of KANs, we propose a novel *Spline-Wise* (SW) symbolic regression algorithm for KAN-based models that systematically converts a trained KAN into a fully symbolic equation. In simple terms, the goal of the SW symbolic regression algorithm is to replace each trained spline activation with a symbolic expression and compose them according to the KAN architecture into a mathematical formula. While drawing inspiration from previous work (Liu et al., 2025b), our procedure incorporates a principled trade-off between expression complexity and accuracy. The procedure is as follows:

1. **Affine Function Fitting.** Given a trained KAN, let $\mathcal{S}$ denote the set of all its spline activations remaining after pruning, and let $\mathcal{O} = \{f_1, \ldots, f_M \mid f_m : \mathbb{R} \to \mathbb{R}\}$, be a fixed library of candidate univariate symbolic primitives (e.g., sin, exp, `square`, `identity`). For each spline $\phi \in \mathcal{S}$, let $\mathbf{x}_\phi = (x_\phi^{(1)}, \ldots, x_\phi^{(N_\phi)}) \in \mathbb{R}^{N_\phi}$ be the vector of pre-activations of $\phi$ collected by performing a forward pass over training data, i.e., the outputs of the KAN neuron in the previous layer that feeds into $\phi$, and let $\boldsymbol{\phi}(\mathbf{x}_\phi) = (\phi(x_\phi^{(1)}), \ldots, \phi(x_\phi^{(N_\phi)}))$ be the corresponding spline outputs. For each candidate $f_m \in \mathcal{O}$, define the affine-transformed candidate

$$\tilde{f}(x; \theta) \ = \ a \cdot f(b \cdot x + c) + d, \qquad \theta = (a, b, c, d) \in \mathbb{R}^4, \tag{8}$$

   and compute the optimal affine parameters $\theta_{f,\phi}^*$ by non-linear least squares:

$$\theta_{f,\phi}^* \ = \ \arg\min_{\theta \in \mathbb{R}^4} \underbrace{\frac{1}{N_\phi} \sum_{s=1}^{N_\phi} \left( \phi\left(x_\phi^{(s)}\right) - \tilde{f}\left(x_\phi^{(s)}; \theta\right) \right)^2}_{\text{MSE}_\phi(f, \theta)}. \tag{9}$$

2. **Complexity-Penalized Function Selection.** For each spline, we must now select the best symbolic representation from the fitted candidates with affine transformation. We search for a function that minimizes a penalized error, balancing approximation accuracy with structural complexity. Specifically, let $\Gamma$ be a range of regularization hyperparameters. For each $\phi \in \mathcal{S}$ and $\gamma \in \Gamma$, define:

$$f_{\phi,\gamma}^* \ = \ \arg\min_{f \in \mathcal{O}} \left[ \text{MSE}_\phi\left(f, \theta_{f,\phi}^*\right) \ + \ \gamma \cdot \text{Complexity}\left(f, \theta_{f,\phi}^*\right) \right], \tag{10}$$

   where $\text{Complexity}(f, \theta_{f,\phi}^*) \in \mathbb{N}$ counts the number of operators in the symbolic expression $\tilde{f}(\cdot; \theta_{f,\phi}^*)$, computed using the `count_ops` method from `sympy` library. The notation $f_{\phi,\gamma}^*$ denotes the candidate expression selected at regularization level $\gamma$ for spline $\phi$; its associated affine parameters are $\theta_{f_{\phi,\gamma}^*,\phi}^*$. For the sake of notation, in what follows we use $f_{\phi,\gamma}^*$ to refer directly to the corresponding affine-transformed function $\tilde{f}(\cdot; \theta_{f_{\phi,\gamma}^*,\phi}^*)$, thereby avoiding the explicit repetition of $\theta_{f_{\phi,\gamma}^*,\phi}^*$.

3. **Pareto-Optimal Formula Selection.** The previous step yields a set of $|\Gamma|$ candidate symbolic functions for each spline, representing a Pareto front of accuracy versus complexity:

$$\mathcal{P}_\phi = \{(c_k, \ell_k)\}_{k=1}^{|\Gamma|}, \quad c_k = \text{Complexity}(f_{\phi,\gamma_k}^*), \quad \ell_k = \log\left[\text{MSE}_\phi\left(f_{\phi,\gamma_k}^*\right)\right], \tag{11}$$

   sorted by increasing complexity $c_k$. The Pareto-optimal function $f_\phi^*$ is selected as the expression with the highest performance-complexity score, defined as the negative (discrete) gradient of the log-MSE with respect to complexity (Cranmer, 2023):

$$f_\phi^* = f_{\phi,\gamma_{k^*}}^*, \quad k^* = \arg\max_{k \in \{1, \ldots, |\Gamma|\}} \begin{cases} 0, & k = 1, \\ -\dfrac{\ell_k - \ell_{k-1}}{c_k - c_{k-1}}, & k = 2, \ldots, |\Gamma|. \end{cases} \tag{12}$$

This corresponds to selecting the expression that achieves the largest reduction in log-MSE relative to the increase in complexity.

4. **Symbolic Model Reconstruction.** Finally, we replace each spline $\phi$ in the trained KANs $\hat{H}$ and $\hat{G}$ with its selected symbolic counterpart $f_\phi^*$. By composing these elementary functions according to the KANs' architectures, we reconstruct the complete symbolic formula $f_{SW} \in \mathcal{F}_{\mathcal{O}}$, following the structure of Equation 5.

The pseudo-code of the above algorithm can be found in Appendix A.4. We design this procedure because the symbolic regression algorithm originally proposed by the KAN authors (Liu et al., 2025b), while accurate, tends to produce expressions of very high structural complexity (see Appendix C.4), undermining the interpretability that motivates using KANs in the first place. On the contrary, our procedure favors expressions that retain the structural insight of the trained network while remaining compact.

### 3.5 Evaluation Pipeline

The ultimate test of a discovered dynamical law is its ability to forecast the system's evolution. Our primary performance measure is, therefore, the MAE between ground-truth trajectories and the predictions obtained by numerically integrating the learned symbolic dynamics. Formally, given a sequence of observations $\{\mathbf{X}(t)\}_{t=0}^{T}$, let $\hat{H}_{SR}$ and $\hat{G}_{SR}$ be the extracted symbolic formulas. Since they describe the structure of an ODE, we can integrate them over any time interval $[t_0, t_m] \subseteq [0, T]$:

$$\hat{\mathbf{x}}_i(t_m) = \mathbf{x}_i(t_0) + \int_{t_0}^{t_m} \left[ \hat{H}_{SR}\big(\hat{\mathbf{x}}_i(t)\big) + \sum_{j=1}^{N} A_{ij}\, \hat{G}_{SR}\big(\hat{\mathbf{x}}_i(t), \hat{\mathbf{x}}_j(t)\big) \right] dt. \tag{13}$$

Our assessment begins with a given set of initial conditions $\mathbf{X}(t_0)$ from a test trajectory, which are then used to integrate the symbolic model via Equation 13 for all subsequent time steps, resulting in a predicted trajectory $\{\hat{\mathbf{X}}(t)\}_{t=t_0+1}^{t_m}$. We then compute the trajectory mean absolute error, $\mathrm{MAE_{traj}}$, between the ground-truth observations and predictions:

$$\mathrm{MAE_{traj}} = \frac{\sum_{i=1}^{N} \sum_{t=t_0+1}^{t_m} |\mathbf{x}_i(t) - \hat{\mathbf{x}}_i(t)|}{N(t_m - t_0 - 1)}. \tag{14}$$

This integration is autoregressive, meaning that prediction errors at one step are propagated into the next. Consequently, even minor inaccuracies in the discovered equations can compound over time, making the $\mathrm{MAE_{traj}}$ a stringent and comprehensive test of a model's long-term accuracy and stability.

A crucial component of our methodology is the rigorous selection of the final symbolic model. Recognizing that models may overfit to a specific network instance, and that both the GP-based and the SW fitting procedures are sensitive to hyperparameters, we design a robust evaluation pipeline to validate SR hyperparameters and test models performance in an Out-Of-Distribution (OOD) setting. These aspects are largely overlooked in previous works, which typically apply SR without systematic hyperparameter validation and evaluate symbolic expressions on trajectories derived from the same graph dynamical system used during training. The proposed pipeline consists of generating an additional validation set by simulating the same dynamics on a *new* graph with a different topology and initial conditions. For each candidate symbolic formula produced by the SR algorithms, we compute the trajectory rollout error (Equation 14) on this *validation set*. The symbolic form achieving the lowest $\mathrm{MAE_{traj}}$ is selected as the definitive expression representing the underlying ODE. Then, we assess the generalization of trained models and extracted equations in a final OOD *test set* that includes three unique simulations, each with distinct graph topology and random initial conditions. We report the $\mathrm{MAE_{traj}}$ averaged over these three test trajectories for both models and formulas, indicating their generalization beyond the training domain. To provide a clear visual guide to our evaluation pipeline, Figure 3 illustrates the overall process for training, symbolic distillation, and evaluation of all models.

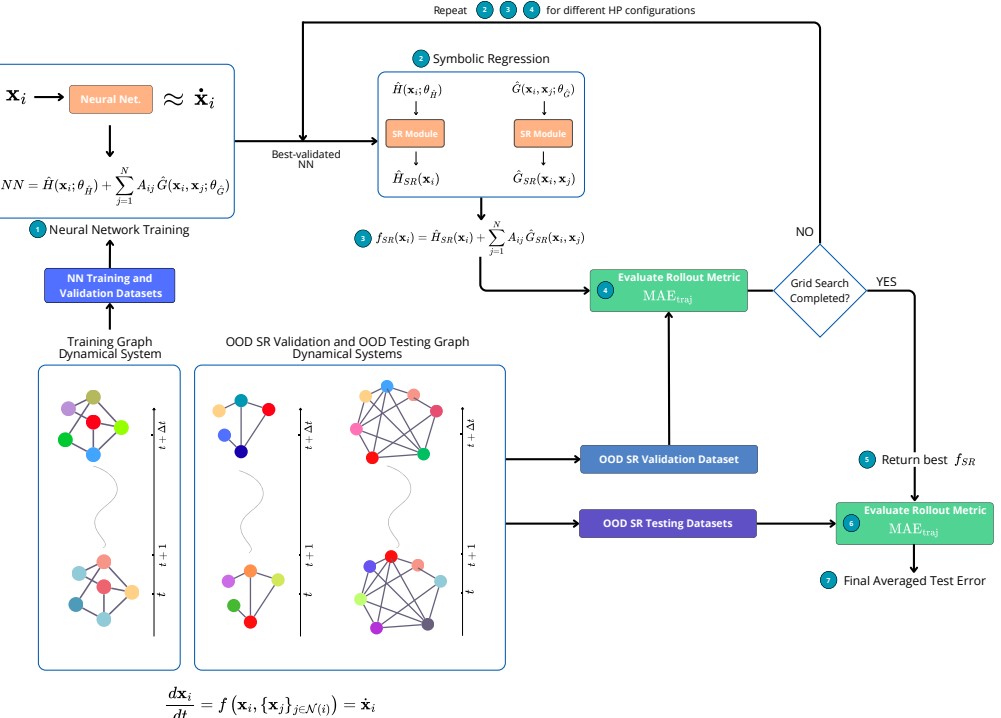

Figure 3: Overview of the experimental pipeline. Starting from a graph dynamical system, we (1) train a neural model on one graph instance, (2) distill symbolic equations via black-box GP or white-box SW regression, (3–4) validate candidates on the SR validation set, and (5–7) evaluate the best formula on OOD test graphs using the long-term $\text{MAE}_{\text{traj}}$ metric.

## 4 Experimental Design

This section outlines the assessed models and the employed datasets. The Appendices and source code offer further information on dataset generation, model implementation, optimization, SR algorithms, and hyperparameters, ensuring scientific reproducibility and fairness.

### 4.1 Models under Assessment

We rigorously and fairly assess a set of distinct state-of-the-art methodologies for inferring the governing equations of dynamical systems on graphs. In addition to the proposed GKAN-ODE model, we test three other approaches: the GMLP-ODE model, the neural architecture of LLC (Hu et al., 2025), and the TPSINDy algorithm (Gao & Yan, 2022). The GMLP-ODE serves as the direct MLP-based counterpart of GKAN-ODE, where the two KANs are replaced by MLPs, allowing for a controlled comparison between the two architectures. LLC is included as a state-of-the-art neural baseline, notable for its MLP-based architecture that explicitly introduces multiplication in the network's structure for $\hat{G}$ in a manner conceptually similar to GKAN-ODE. Unlike these neural approaches, TPSINDy directly learns sparse symbolic expressions for $\hat{H}$ and $\hat{G}$ from data and represents the leading non-neural approach. For the neural architectures, we utilize SR procedures to extract interpretable equations: as a black-box SR, the GP-based tool PySR (Cranmer, 2023) is employed, and the resulting symbolic models are labeled with the suffix "+GP"; similarly, the SW fitting applied to our proposed model is referred to as GKAN-ODE+SW. Finally, the complexity of the retrieved formulas is computed using the `count_ops` method from the `sympy` library, which counts the number of mathematical operations in a given mathematical expression.

## 4.2 Inference on Synthetic Dynamical Systems

We first evaluate the models' capacity to recover the precise symbolic form of known dynamics. To this end, we utilize four canonical network dynamical systems, chosen to represent a diverse range of nonlinearities common in scientific models (Barzon et al., 2024): Kuramoto oscillators (KUR), epidemic spreading (EPID), biochemical (BIO), and population (POP) dynamics. We generate these synthetic datasets by integrating the models on a fixed Barabási–Albert (Barabási & Albert, 1999) network, chosen for its scale-free topology, which is representative of many real-world systems. To evaluate robustness against measurement uncertainty, we also create noisy variants of these systems by adding white noise to node states at each time step under different signal-to-noise ratio (SNR) levels. For all experiments with this setting, models are trained on the first 80% of the temporal observations, with the remaining 20% reserved for validation and hyperparameter tuning. The ground truth equations of the considered dynamics are reported in Table 1, while additional details on dataset creation are provided in Appendix B.1. The evaluation of the trained neural models and their respective distilled formulas is performed according to the evaluation pipeline described in Section 3.5.

## 4.3 Inference on Real-World Empirical Data

To assess performance on a task with unknown ground truth, we utilize the empirical dataset of epidemic dynamics from Gao & Yan (2022), which captures the early pre-intervention spread of the H1N1, SARS, and COVID-19 outbreaks across the global airline network. We train the neural models on the COVID-19 dataset and extract symbolic representations. As a true OOD validation set is unavailable, we select the symbolic expression that yields the lowest $\text{MAE}_{\text{traj}}$ on the validation data extracted from the training set. This procedure discovers a single homogeneous equation describing the global average dynamics. To account for country-specific variations, we then fine-tune the coefficients of this discovered symbolic structure for each node, following the ideas proposed in previous works (Gao & Yan, 2022; Hu et al., 2025), and detailed in Appendix D.1. Since our evaluation focuses on the generalizability of the discovered laws, we investigate whether the symbolic structures learned from COVID-19, with only coefficient fine-tuning, can effectively model H1N1 and SARS outbreaks. For the final model assessment, we primarily evaluate predictive performance using the long-term trajectory rollout metric, $\text{MAE}_{\text{traj}}$. In addition, for comparison with previous studies (Gao & Yan, 2022), we report the short-term, single-step prediction metric, $\text{MAE}_{\text{eul}}$. This metric relies on predicted trajectories computed using an Euler integration scheme that performs one-step-ahead forecasts using ground truth data at each step, rather than the previously predicted state. As a result, it reflects short-term fitting accuracy, but does not constitute a proper evaluation of long-term trajectory prediction error, since it avoids the error accumulation over time.

# 5 Results and Discussion

## 5.1 Comparative Performance on Synthetic Systems

Our first key finding, illustrated in Figure 4 (left), is the superior performance of neural-based architectures over the sparse regression baseline, TPSINDy. The neural models, both before and after symbolic distillation, consistently yield more accurate and stable long-term trajectory rollouts, as measured by the $\text{MAE}_{\text{traj}}$. TPSINDy correctly identifies the KUR dynamics, arguably due to its expressiveness in periodic functions; however, it fails in the other cases. Notably, on EPID and POP dynamics, TPSINDy suffers from catastrophic error accumulation, leading to a high $\text{MAE}_{\text{traj}}$. This limitation likely stems from its reliance on a pre-defined library of candidate functions: while it may perform well in certain cases, it also prevents it from discovering composite or nested functional forms not explicitly inserted in the initial libraries. Conversely, neural-based approaches are universal approximators without this restriction, and the symbolic distillation through SR can easily compose complex nested equations starting from a restricted library of univariate functions and simple binary operators. Furthermore, the poor performance of TPSINDy on BIO dynamics shows that having all the necessary terms in the initial functional libraries is not a guarantee for the model to learn the correct expression. Indeed, although the BIO dynamics can in principle be expressed as a linear combination of non-linear terms already present in TPSINDy libraries, the model fails in identifying the structure of the ground-truth dynamics (as shown in Table 2).

Among the neural approaches, the models derived from GKAN-ODE architecture achieve the lowest trajectory errors across all four systems, with the black-box symbolic model (GKAN-ODE+GP) performing best. The LLC architecture also performs well, particularly compared to its GMLP-ODE counterpart. Beyond raw performance, GKAN-ODE models are also more parameter-efficient than the other neural-based models: Figure 4 (right) provides a clear visualization of the trade-off between performance (MAE$_{\text{traj}}$) and the number of parameters of each model.

As a further validation, the topology-agnostic MLP-ODE baseline (Appendix C.5) suffers catastrophic error accumulation on three of four systems, confirming that network structure is essential for accurate equation discovery. Moreover, Appendix C.7 presents an ablation study on derivative estimation methods, while Appendix C.2 provides additional trajectory metrics, confirming the stability of the observed performance trends.

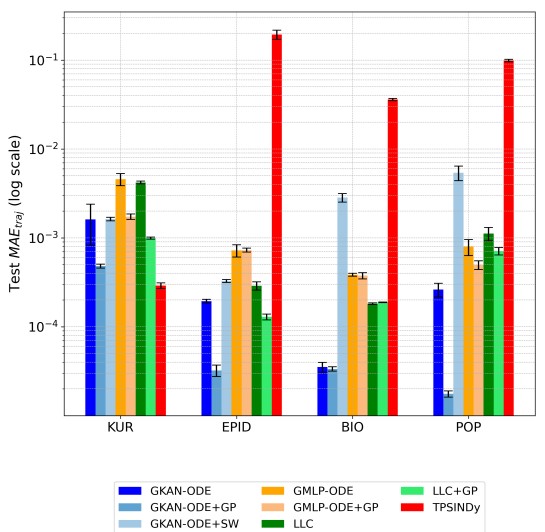
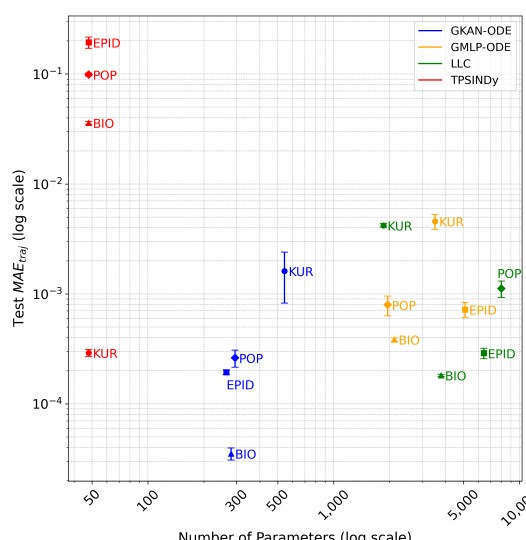

Figure 4: Performance comparison on synthetic dynamics. (Left) Comparison of test MAE$_{\text{traj}}$ for both models and the inferred equations. (Right) Test MAE$_{\text{traj}}$ and number of parameters of the trained neural-based models and TPSINDy (whose parameters are defined by its symbolic function library). Values are averaged on three test graphs and the standard deviation is reported as error bars.

## 5.2 Symbolic Discovery and Interpretability

As shown in Table 2, the black-box GP procedure for extracting equations from neural models recovers the exact symbolic form of the governing equations (Table 1) for all four synthetic systems, up to algebraic transformations. Additionally, the fitted coefficients closely match the ground truth (within 0.1% relative error), supporting the effectiveness of the pipeline from neural training to symbolic distillation. Although the formulas extracted from the neural models are similar to each other, the GKAN-ODE+GP expressions yield coefficients closer to the ground truth, leading to a lower accumulated error over time compared to equations with coefficients that deviate more from the ground truth. This behavior is further inspected in Appendix C.1, where we present the temporal evolution of MAE$_{\text{traj}}$ for the neural models.

The structure-aware GKAN-ODE+SW approach offers a more direct window into the model's inner workings. Detailed in Table 3, this method also successfully identifies the correct underlying dynamics. For the BIO system, it retrieves slightly different coefficients, while for the KUR system, it discovers a phase-shifted sine function that is mathematically equivalent to the ground truth. For EPID and POP dynamics, the SW method yields expressions with additional small-coefficient terms or minor parameter deviations (e.g., $< 2\%$). While numerically small, these deviations can accumulate during autoregressive integration, leading to a higher MAE$_{\text{traj}}$. While the GP approach imposes a stronger global simplicity prior to finding the most

compact formula, the SW approach provides a more granular representation of what the individual splines have learned from the data, including nuances that are propagated in the construction of the final formula starting from individual spline fittings. However, as discussed in Appendix C.4, our proposed SW algorithm retrieves more compact symbolic formulas than the ones obtained by the original KAN symbolic regression method, while maintaining comparable performance.

Table 1: Ground-truth governing equations for the four synthetic dynamical systems.

| Dataset | Ground-Truth Equation |
|---|---|
| KUR | $2 + \frac{1}{2} \sum_j A_{ij} \sin(x_j - x_i)$ |
| EPID | $-\frac{1}{2} x_i + \frac{1}{2} \sum_j A_{ij}(1 - x_i) x_j$ |
| BIO | $1 - \frac{1}{2} x_i - \frac{1}{2} \sum_j A_{ij} x_i x_j$ |
| POP | $-\frac{1}{2} x_i + \sum_j A_{ij} \frac{x_j^3}{5}$ |

Table 2: Learned symbolic expressions and their complexities across models and synthetic datasets.

| Model | Dataset | Learned Expression | Complexity |
|---|---|---|---|
| GKAN-ODE + GP | KUR | $1.9992 + \sum_j A_{ij}(-0.5005 \cdot \sin(x_i - x_j))$ | 5 |
| | EPID | $-0.4997 \cdot x_i + \sum_j A_{ij}(x_j \cdot (0.5001 - 0.5002 \cdot x_i))$ | 6 |
| | BIO | $-0.5006 \cdot x_i + 1.0002 + \sum_j A_{ij}(-0.4998 \cdot x_i x_j)$ | 6 |
| | POP | $-0.4999 \cdot x_i + \sum_j A_{ij}(0.2000 \cdot x_j^3)$ | 5 |
| GMLP-ODE + GP | KUR | $2.0009 + \sum_j A_{i,j}(-0.4971 \cdot \sin(x_i - x_j))$ | 5 |
| | EPID | $-0.4990 \cdot x_i + \sum_j A_{i,j}(0.4976 \cdot x_j \cdot (1.0000 - x_i))$ | 6 |
| | BIO | $-0.4970 \cdot x_i + 0.9987 + \sum_j A_{i,j}(-0.4989 \cdot x_i x_j)$ | 6 |
| | POP | $-0.4998 \cdot x_i + \sum_j A_{i,j}(0.1973 \cdot x_j^3)$ | 5 |
| LLC + GP | KUR | $1.9995 + \sum_j(-0.4986 \cdot \sin(x_i - x_j))$ | 5 |
| | EPID | $-0.5012 \cdot x_i + \sum_j A_{i,j}(x_j \cdot (0.5005 - 0.5003 \cdot x_i))$ | 6 |
| | BIO | $-0.4971 \cdot x_i + 0.9977 + \sum_j A_{i,j}(-0.4992 \cdot x_i x_j)$ | 6 |
| | POP | $-0.4973 \cdot x_i + \sum_j A_{i,j}(0.1962 \cdot x_j^3)$ | 5 |
| TPSINDy | KUR | $2.0000 + \sum_j A_{i,j}(0.4994 \cdot \sin(x_j - x_i))$ | 5 |
| | EPID | $-0.5679 + \sum_j A_{i,j}(0.2084 \cdot \exp(x_j - x_i))$ | 4 |
| | BIO | $0.8670 + \sum_j A_{i,j}(-0.7113 \cdot x_i x_j)$ | 4 |
| | POP | $-0.0162 + \sum_j A_{i,j}(0.0400 \cdot x_j + 0.0031 \cdot \sin(x_j))$ | 5 |

Table 3: Best-validated Spline-wise symbolic formulas $f_{SW}$ and their structural complexity for the GKAN-ODE+SW model on the four synthetic dynamical systems.

| Dataset | GKAN-ODE+SW Discovered Symbolic Expressions | Complexity |
|---|---|---|
| KUR | $1.9991 + \sum_j A_{ij}(-0.5005 \cdot sin(-0.9992 \cdot x_i + 0.9995 \cdot x_j + 3.1373))$ | 8 |
| EPID | $-0.4988 \cdot x_i + \sum_j A_{ij}(-0.4961 \cdot x_i x_j + 0.4970 \cdot x_j - 0.0022 \cdot x_i + 0.0018)$ | 10 |
| BIO | $-0.5000 \cdot x_i + 1.0001 + \sum_j A_{ij}(-0.4899 \cdot x_i x_j)$ | 6 |
| POP | $-0.2862 x_i - 0.1744 \tanh(1.4270 x_i - 0.0779) - 0.0122$ $+ \sum_j A_{ij}(0.1474 x_j^3 + 0.0066 x_j^2 + 0.0204 x_j)$ | 16 |

## 5.3  Observational Noise Scenario

In the data-generating process, independent Gaussian noise is added to the node states at each time step under three different signal-to-noise ratio (SNR) levels expressed in decibels (dB): 70 dB, 50 dB, and 20 dB. The performance of the extracted symbolic expressions in this setting is depicted in Figure 5. The methods are robust to noise up to 50 dB of SNR, particularly neural models, even though the quality of the expressions degrades with increasing levels of noise. This degradation is further exacerbated by the fact that we estimate numerical derivatives, which are highly sensitive to noise and can amplify small fluctuations in the data. To mitigate this effect, we also evaluate the use of a denoising algorithm prior to derivative estimation, with the corresponding results reported in Appendix C.6.

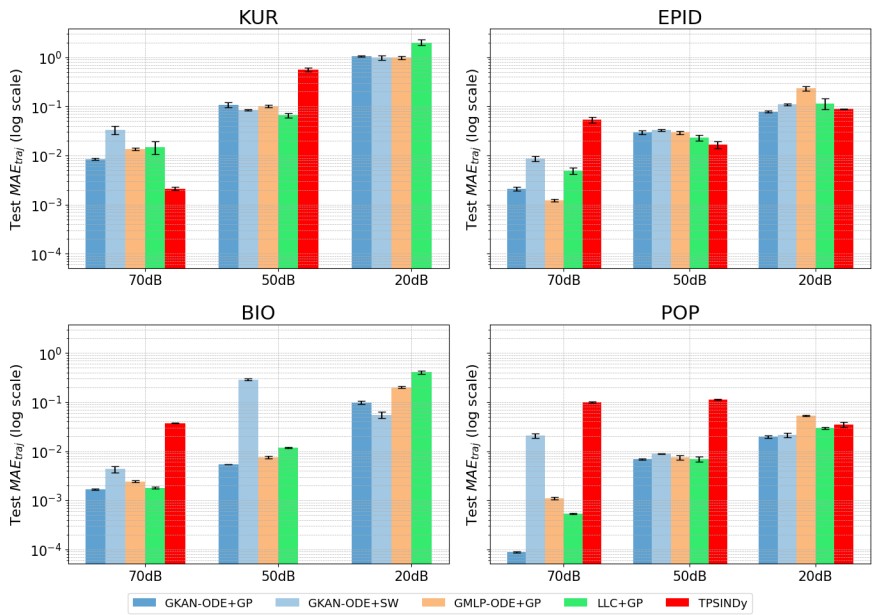

Figure 5: Performance of the extracted symbolic expression across various levels of SNR for each synthetic dataset. Missing values of TPSINDy are due to numerical divergences.

## 5.4  Discovery in Real-World Epidemic Dynamics

Under this scenario, the symbolic global structures of the learned equations are reported in Table 4. The

Table 4: Symbolic expressions extracted from COVID-19 data as global dynamics, before country-specific fine-tuning. The LLC equation is re-derived from scratch, as the original work does not report all necessary coefficients needed for reproduction.

| Model | Discovered Symbolic Expression | Complexity |
|---|---|---|
| TPSINDy | $a \cdot x_i + b \cdot \sum_j A_{ij}\, 1/(1 + e^{-(x_j - x_i)})$ | 7 |
| LLC+GP | $a \cdot \tanh(x_i + b) + c \cdot \sum_j A_{ij}((x_i - x_j) \cdot e^{-x_j})$ | 7 |
| GMLP-ODE+GP | $a \cdot \ln(x_i + b) + \sum_j A_{ij} \ln(\tan(x_i + c)^2 + d)$ | 9 |
| GKAN-ODE+GP | $ax_i + b + \sum_j A_{ij}\left(c \cdot e^{x_j}\right)$ | 5 |
| GKAN-ODE+SW | $a \cdot \tanh\big(b \cdot \tanh(cx_i + d) + e\big) - f \cdot \tanh\big(gx_i^3 + hx_i^2 - ix_i - j\big) + k$ $+ \sum_j A_{ij}\big(l \cdot \tanh(mx_i - n) - o \cdot \tanh(px_j - q) - r\big)$ | 30 |

models yield diverse functional forms, with TPSINDy favoring a logistic-like interaction, while neural architectures learn more complex nonlinearities. This scenario further highlights the critical trade-off between

model expressivity and interpretability. Notably, the GKAN-ODE+GP model distills a particularly simple and plausible law, suggesting a linear self-term with an exponential growth interaction from neighbors. In contrast, the GKAN-ODE+SW method produces a significantly more complex expression by directly translating the KAN's internal splines. This presents a choice for domain experts: pursuing the simplest explanatory model (via GP) or analyzing a more complex representation of the neural-learned dynamics (via SW). Despite the SW expression being very complex, it achieves comparable performance to its GP counterpart, indicating that the SW algorithm is correctly distilling a richer, albeit less interpretable, equation.

Key evaluation rests on the models' stability over time and their generalization to unseen data. Figure 6 contrasts the performance of the discovered equations over the three epidemic dynamics after tuning the coefficients, assessing the expressions' adaptability to varying epidemiological scenarios. While TPSINDy is competitive in single-step forecasting ($\mathrm{MAE_{eul}}$), it leads to catastrophic error accumulation in long-term trajectory rollouts ($\mathrm{MAE_{traj}}$). Conversely, all neural-derived laws exhibit both strong long-term stability and short-term predictive accuracy. In Figures 7 and 8, we show the performance of the learned symbolic expressions on COVID-19 data for a subset of 4 countries. Figure 7 presents the predicted trajectories obtained through autoregressive integration, while Figure 8 illustrates the results from short-term integration. As suggested by the performance comparison depicted in Figure 6, neural-based models are able to capture the epidemiological spreading in both scenarios, while TPSINDy struggles in long-term predictions.

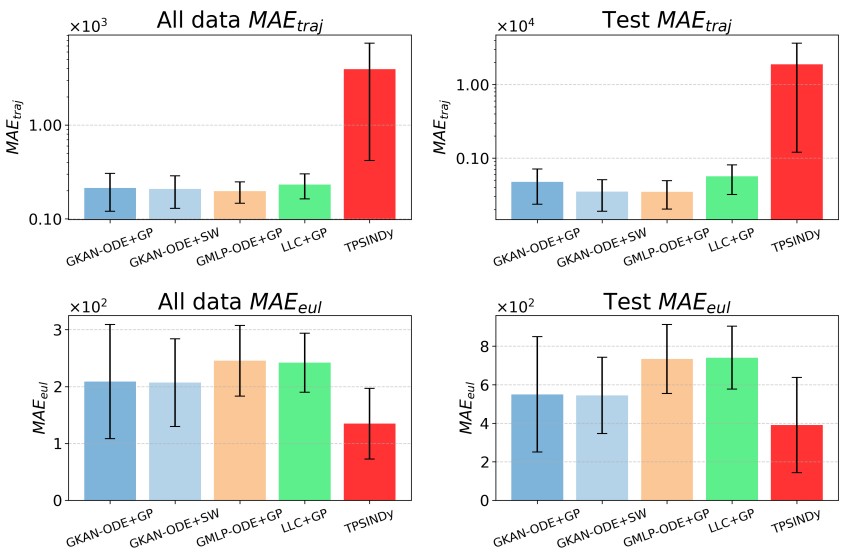

Figure 6: Performance comparison of the symbolic formulas of Table 4 averaged on the COVID, H1N1, SARS datasets. Both $\mathrm{MAE_{traj}}$ (top) and $\mathrm{MAE_{eul}}$ (bottom) are computed on the complete (left) and test (right) datasets. The test sets consist of the last 10% of observations of each dataset.

# 6 Conclusion

This paper rigorously assesses the most prominent AI methods of equation discovery for graph dynamical systems to reveal their true performance. Given the limited number of existing methods for symbolic equation discovery on graphs, the evaluation pipeline and open-source codebase introduced here are designed to serve as a foundation for future research in this area, enabling fair comparison of new architectures and symbolic distillation strategies as they emerge. Our findings establish that neural-based approaches consistently outperform the sparse regression baseline, and that GKAN-ODE models demonstrate a superior balance of predictive accuracy, parameter efficiency, and an architecture inherently amenable to interpretation.

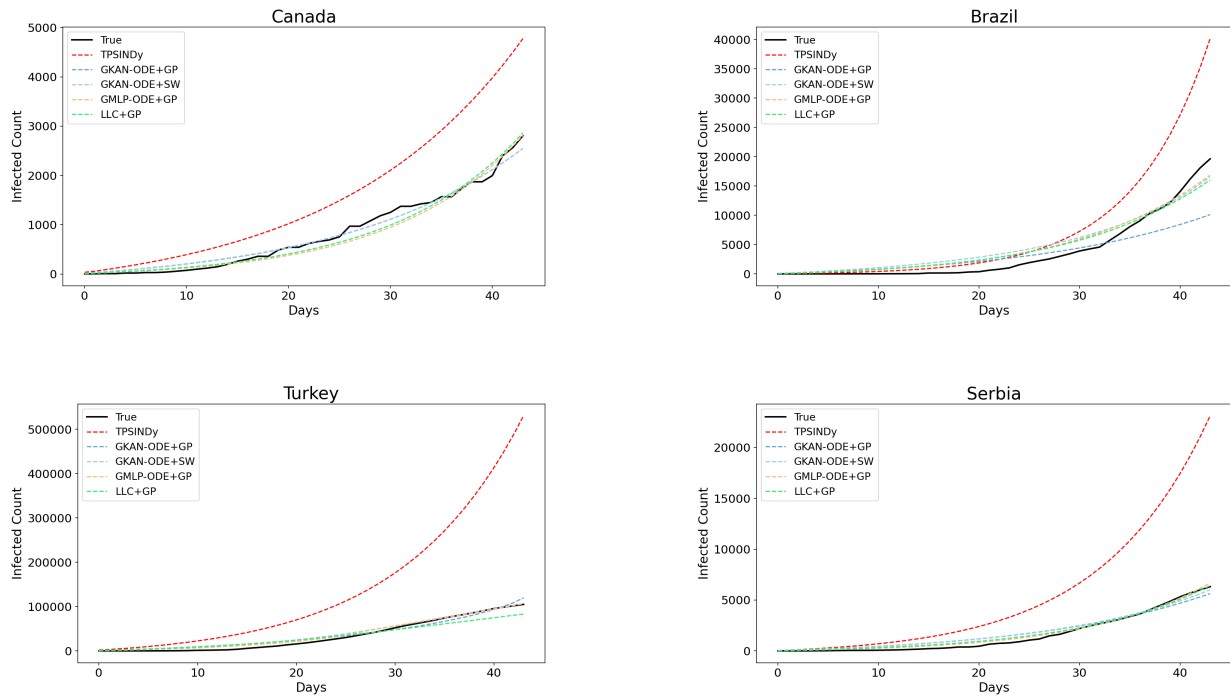

Figure 7: Predicted trajectories obtained by the long term (autoregressive) integration of the learned equations on COVID-19 data of Canada, Brazil, Turkey and Serbia.

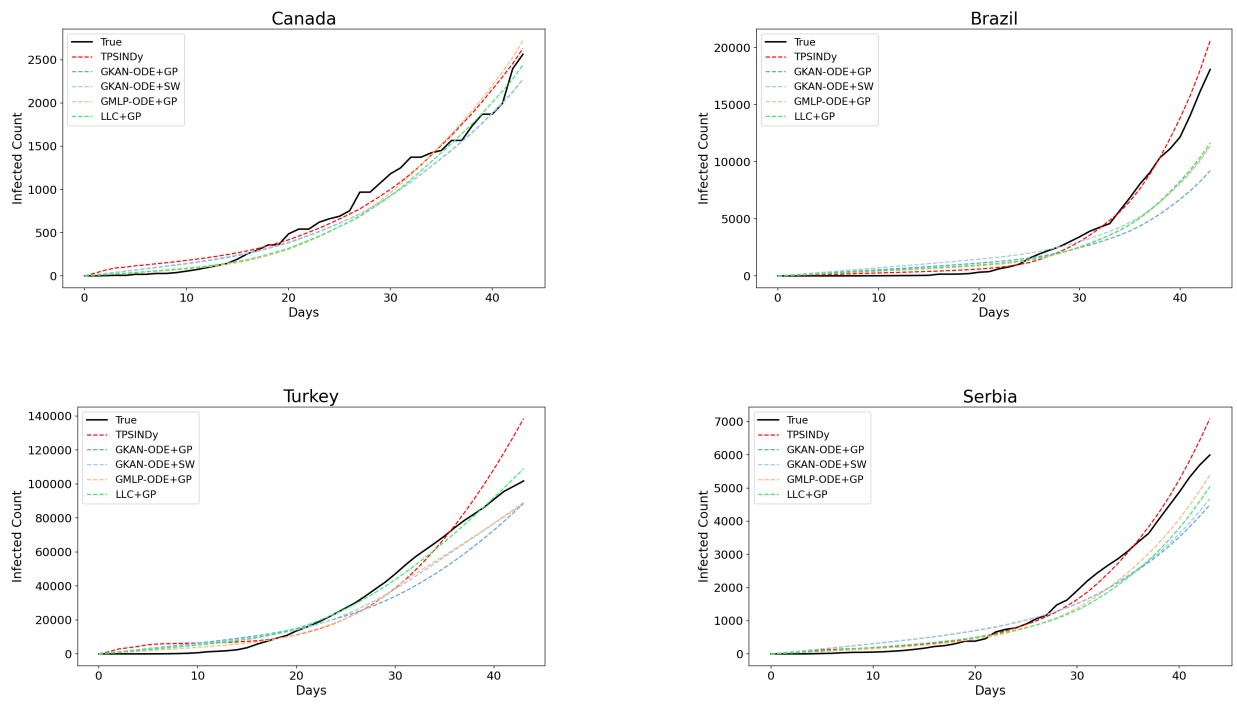

Figure 8: Predicted trajectories obtained by the short term integration of the learned equations on COVID-19 data of Canada, Brazil, Turkey and Serbia.

A key contribution of this work lies in the distillation of symbolic knowledge from these models. We have shown how a model-agnostic SR method can effectively recover the ground-truth equations. In parallel, our novel Spline-Wise fitting algorithm provides a transparent and truthful, albeit more granular, symbolic representation of KANs' internal logic. Moreover, the SW algorithm paired with the proposed KAN architecture, incorporating multiplicative nodes, achieves a superior complexity–performance trade-off compared to the original KAN SW fitting approach.

By establishing a reproducible pipeline and advocating for evaluation based on long-term, out-of-distribution generalization, this work aims to serve as a practical reference and open-source contribution for the interdisciplinary scientific community working on complex systems. It clarifies the state-of-the-art and promotes a human-in-the-loop paradigm where AI acts as a powerful collaborator that generates plausible, testable hypotheses, thereby augmenting human intuition and understanding.

**Limitations.** Our framework assumes a static graph topology and autonomous (time-invariant) ODEs, following previous works (Gao & Yan, 2022; Hu et al., 2025). The extension to time-varying topologies or non-autonomous dynamics remains open. The Spline-Wise algorithm, while transparent, produces more complex expressions than the black-box GP approach, and its accuracy degrades for systems with strongly multiplicative interactions (e.g., POP). Furthermore, all methods rely on numerical derivative estimation, which is sensitive to observation noise; our denoising analysis (Appendix C.6) partially mitigates but does not eliminate this limitation. Finally, our experiments are limited to scalar node states ($d = 1$); scaling to multivariate node features warrants further investigation.

Ultimately, this study demonstrates that interpretable neural architectures, particularly KANs, are effective tools for discovering governing equations of graph dynamical systems. We anticipate that combining the structural transparency of KANs with principled symbolic distillation will extend naturally to PDEs, multivariate dynamics, and time-evolving network topologies as well.

### Acknowledgments

We acknowledge the financial support of the Italian Ministry of University and Research (MUR) under the PRIN program – Progetti di Rilevante Interesse Nazionale – PRIN 2022 (Secretary-General's Decree No. 1401 of 18/09/2024) – CUP C53C24000770006, project title "DEEP-GRAPH: Design and Theory of Deep Graph Learning".

This work was also supported by the National Recovery and Resilience Plan (NRRP) Mission 4 Component 2 Investment 1.3 - Call for tender No. 341 of March 15, 2022, from the Italian Ministry of University and Research - NextGenerationEU; project code PE0000013, Concession Decree No. 1555 of October 11, 2022, CUP C63C22000770006, project title "Future AI Research (FAIR) - Spoke 2 Integrative AI - Symbolic conditioning of Graph Generative Models (SymboliG)".

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

## Appendix

## A  Methodological Details

### A.1  KAN Sparsity Loss Function

KANs are usually trained with a sparsity loss, which is an adaptation of the $L^1$ norm of MLPs. However, this norm is directly defined on the learned activation functions. Formally, the $L^1$ norm of an activation function $\phi$ is given by the average magnitude over its $N_p$ inputs, that is:

$$|\phi|_1 \equiv \frac{1}{N_p} \sum_{s=1}^{N_p} |\phi(x^{(s)})|. \tag{15}$$

Then, the $L^1$ norm of a KAN layer $\boldsymbol{\Phi}$ with $d_{in}$ inputs and $d_{out}$ outputs is defined as:

$$|\boldsymbol{\Phi}|_1 \equiv \sum_{i=1}^{d_{in}} \sum_{j=1}^{d_{out}} |\phi_{i,j}|_1, \tag{16}$$

that is, the sum of $L^1$ norms of all the activation functions in $\boldsymbol{\Phi}$. Furthermore, an entropy term is added to the loss definition:

$$S(\boldsymbol{\Phi}) \equiv -\sum_{i=1}^{d_{in}}\sum_{j=1}^{d_{out}} \frac{|\phi_{i,j}|_1}{|\boldsymbol{\Phi}|_1} log\left(\frac{|\phi_{i,j}|_1}{|\boldsymbol{\Phi}|_1}\right). \tag{17}$$

Then, the final training loss $\mathcal{L}_{total}$ is given by the prediction loss $\mathcal{L}_{pred}$ plus the $L^1$ and entropy regularization aggregated over all the layers:

$$\mathcal{L}_{total} = \mathcal{L}_{train} + \lambda\left(\mu_1 \sum_{l=1}^{L}|\boldsymbol{\Phi}_l|_1 + \mu_2 \sum_{l=1}^{L}S(\boldsymbol{\Phi}_l)\right), \tag{18}$$

where $\mu_1, \mu_2$, and $\lambda$ are hyperparameters that determine the impact of the corresponding loss terms.

One of the key characteristics of KANs is that they can be used to perform symbolic regression. Specifically, once the model is trained, it is possible to prune inactive neurons by looking at their spline activation magnitudes and then fixing the remaining activation functions to symbolic formulas (e.g., `sin`, `cos`) so that the whole model can be described through a symbolic representation. This process enhances interpretability, as it overcomes the black-box nature typical of deep-learning models by providing, as output, a human-readable mathematical formulation of the learned function. Refer to the original paper (Liu et al., 2025b) for additional details on the pruning and symbolic regression procedures.

## A.2   Technical Implementation

We implemented the model under consideration using Python 3.12.0, Pytorch 2.3.1, and PyG 2.3.1. For hyper-parameter tuning, we employed the Optuna package (Version 4.3.0). The Spline-Wise fitting procedure relies on the `curve_fit` method from `scipy` library for solving the non-linear least squares problem, while for the GP-based SR algorithm, we used PySR 1.5.5. We utilized the `dopri5` solver from the `torchdiffeq` library as a numerical integrator for computing the rollout metric $MAE_{traj}$, setting `atol = rtol = `$10^{-5}$ for all models.

## A.3   Hardware Setup

We carried out the experiments on a Google Cloud `n1-standard-16` virtual machine, equipped with 48 vCPUs based on the Intel Cascade Lake CPU architecture and 192 GB of system memory. The setup was further accelerated by 4 NVIDIA L4 GPUs.

## A.4   Spline-Wise Symbolic Regression Algorithm

Algorithm 1 describes the proposed Spline-Wise symbolic fitting procedure of KAN-based models. To ensure parsimony, in line 9, the coefficients with magnitudes below a threshold ($\epsilon$) are pruned before complexity is computed. For example, the expression $x^3 + 10^{-5}x^2$ is considered to have a complexity of 1, not 4. As a measure of complexity, we use the `count_ops` function from `sympy` library, which measures the number of operations an expression contains.

## A.5   Hyperparameters Specifications

This section lists the search spaces of the employed hyperparameters in the experimental analysis. Model selection is performed using the Optuna package, optimizing the MAE over 35 trials for synthetic dynamics, over 70 trials for dynamics with noise, and over 100 trials for COVID-19 data. All neural architectures are optimized using Adam for 1000 epochs with early stopping and patience parameters of 200 for synthetic dynamics and 300 for the real-world COVID dataset. Tables 5, 6 and 7 detail the hyperparameter ranges used for the neural-based models. For TPSINDy, we consider the default libraries of symbolic functions provided by the authors in the original implementation, including polynomial, trigonometric, fractional, and exponential terms.

---

**Algorithm 1** Spline-wise Symbolic Regression for KAN

---

**Require:** Splines $\mathcal{S}$, function library $\mathcal{O}$, regularization grid $\Gamma$, coefficient pruning threshold $\varepsilon$, spline pruning threshold $\rho$, `model_selection`

**Ensure:** Selected symbolic function $f_\phi^*$ for each $\phi \in \mathcal{S}$ and final symbolic formula $f_{SW}$

1: $\mathcal{S}_{\text{pruned}} = \text{pruning}(\mathcal{S}, \rho)$
2: **for** each spline $\phi \in \mathcal{S}_{\text{pruned}}$ **do**
3:      Collect pre-activations $\mathbf{x}_\phi = (x_\phi^{(1)}, \ldots, x_\phi^{(N_\phi)})$ and spline outputs $\phi(\mathbf{x}_\phi)$
4:      $\mathcal{P}_\phi \leftarrow [\,]$
5:      **for** each $\gamma \in \Gamma$ **do**
6:          $f_{\phi,\gamma}^* \leftarrow \varnothing, \quad L_{\min} \leftarrow \infty$
7:          **for** each candidate function $f \in \mathcal{O}$ **do**
8:              Fit affine parameters $\theta_{f,\phi}^* = (a, b, c, d)$ by non-linear least squares
9:              Set to zero the entries of $\theta_{f,\phi}^*$ with magnitude $< \varepsilon$
10:             $\hat{\mathbf{y}}_{f,\phi} \leftarrow \tilde{f}(\mathbf{x}_\phi; \theta_{f,\phi}^*)$
11:             $\text{mse} \leftarrow \text{MSE}_\phi(f, \theta_{f,\phi}^*) = \frac{1}{N_\phi} \sum_{s=1}^{N_\phi} \left(\phi(x_\phi^{(s)}) - \hat{y}_{f,\phi}^{(s)}\right)^2$
12:             $c \leftarrow \text{Complexity}(f, \theta_{f,\phi}^*)$
13:             $L \leftarrow \text{mse} + \gamma \cdot c$
14:             **if** $L < L_{\min}$ **then**
15:                 $L_{\min} \leftarrow L$
16:                 $f_{\phi,\gamma}^* \leftarrow \tilde{f}(\cdot\,; \theta_{f,\phi}^*), \quad c_\gamma \leftarrow c, \quad \ell_\gamma \leftarrow \log(\text{mse})$
17:             **end if**
18:          **end for**
19:          Append $(f_{\phi,\gamma}^*, c_\gamma, \ell_\gamma)$ to $\mathcal{P}_\phi$
20:      **end for**

21:      Sort $\mathcal{P}_\phi$ by increasing complexity, giving $\{(c_k, \ell_k)\}_{k=1}^{|\Gamma|}$ with functions $\{f_{\phi,\gamma_k}^*\}_{k=1}^{|\Gamma|}$
22:      **if** `model_selection` = Score **then**         ▷ favour parsimony: highest performance-complexity score
23:          $\text{score}_1 \leftarrow 0$
24:          **for** $k = 2$ to $|\Gamma|$ **do**
25:              $\text{score}_k \leftarrow -(\ell_k - \ell_{k-1})/(c_k - c_{k-1})$
26:          **end for**
27:          $k^* \leftarrow \arg\max_{k \in \{1,\ldots,|\Gamma|\}} \text{score}_k$
28:      **else**                        ▷ `model_selection` = Log Loss: favour fit, lowest log-MSE
29:          $k^* \leftarrow \arg\min_{k \in \{1,\ldots,|\Gamma|\}} \ell_k$
30:      **end if**
31:      $f_\phi^* \leftarrow f_{\phi,\gamma_{k^*}}^*$
32: **end for**
33: Compose $\{f_\phi^*\}_{\phi \in \mathcal{S}}$ according to the additive/multiplicative structure of the KANs $\hat{H}$, $\hat{G}$ to reconstruct $f_{SW}$
34: **return** $\{f_\phi^*\}_{\phi \in \mathcal{S}}, \ f_{SW} \in \mathcal{F}_\mathcal{O}$

---

Model selection of GP-based and Spline-Wise SR methods is performed via grid search according to the hyperparameter grids specified in Tables 8, 9, respectively.

## B   Experimental Setup and Datasets

### B.1   Synthetic Dataset Generation and Reproducibility Protocols

Table 10 shows the general equations of the studied synthetic dynamical systems, while Table 1 reports the considered instantiations. Preliminary experiments with different dynamic parameters—provided they are physically consistent—did not yield significant differences in terms of the models' learning capabilities; hence,

Table 5: Hyperparameter ranges of the MLPs in the GMLP-ODE models

| Hyperparameter | Values |
|---|---|
| Hidden dimensions | $[8, 64]$ |
| Activation function | $\{\texttt{relu}, \texttt{softplus}, \texttt{tanh}\}$ |
| Dropout probability | $[0.0001, 0.5]$ |
| Hidden layers | $\{1, 2\}$ |
| Learning rate | $[0.0005, 0.05]$ |
| Batch size | $\{16, 32, 64\}$ |

Table 6: Hyperparameter ranges of the MLPs in the LLC models, where $\lambda$ denotes the regularization parameter of the penalized loss function minimized during training.

| Hyperparameter | Values |
|---|---|
| Hidden dimensions | $[8, 64]$ |
| Activation function | $\{\texttt{relu}, \texttt{softplus}, \texttt{tanh}\}$ |
| Hidden layers | $\{1, 2\}$ |
| Learning rate | $[0.0005, 0.05]$ |
| Batch size | $\{16, 32, 64\}$ |
| Regularization $\lambda$ | $[0.0, 0.01]$ |

Table 7: Hyperparameter ranges of the KANs in the GKAN-ODE models.

| Hyperparameter | Values |
|---|---|
| Grid size | $[5, 20]$ |
| Spline order | $[1, 3]$ |
| Range limit | $[-10, 10]$ |
| Hidden dimensions | $[1, 6]$ |
| Regularization $\lambda$ | $[10^{-6}, 1.0]$ |
| Learning rate | $[0.0005, 0.05]$ |
| $\mu_1$ | $[0.1, 1.0]$ |
| $\mu_2$ | $[0.1, 1.0]$ |
| Batch size | $\{16, 32, 64\}$ |

Table 8: Hyperparameter grid for the PySR algorithm.

| Hyperparameter | Values |
|---|---|
| Number of iterations | $\{50, 100, 200\}$ |
| Model selection | $\{\text{Score}, \text{Accuracy}\}$ |
| Binary operators | $[+, -, *, /]$ |
| Unary operators | $[\texttt{exp, sin, neg, square, cube, abs, tan, tanh, ln,}$ $\texttt{zero}]$ |

Table 9: Hyperparameter grid of the Spline-wise fitting algorithm. The *Model selection* parameter is used at line 22 of Algorithm 1, and defines whether to choose the function with the highest score (thus favoring simpler equations) or with the lowest log loss (thus favoring accuracy).

| Hyperparameter | Values |
|---|---|
| Spline pruning threshold $\rho$ | $\{0.01, 0.05, 0.1\}$ |
| Coefficient pruning threshold $\epsilon$ | $\{0.001, 0.01, 0.1\}$ |
| Model selection | $\{\text{Score}, \text{Log loss}\}$ |
| $\Gamma$ | $[10^{-5}, 10^{-4}, 10^{-2}, 10^{-1}, 1]$ |
| Symbolic library $\mathcal{O}$ | `[identity, square, cube, exp, abs, sin, cos, tan, tanh, ln, zero ]` |

we set their magnitude to plausible values considered in the literature (Barzon et al., 2024; Gao & Yan, 2022). The datasets are generated by numerically integrating these models with the Runge–Kutta method of order 5, implemented in the `solve_ivp` function from the `scipy` library, using absolute and relative tolerances of $10^{-12}$ to ensure high numerical precision. We simulate the dynamics on a Barabási–Albert (Barabási & Albert, 1999) graph with 70 nodes and an attachment parameter $m = 3$, saving the solutions at $T = 2000$ regularly spaced time steps. We report in Table 11 the set of parameters to reproduce dataset generation.

Table 10: General equations of the considered dynamical processes.

| Dynamics | Equation $dx_i/dt$ |
|---|---|
| KUR | $\omega + K \sum_j A_{ij} \sin(x_j - x_i)$ |
| EPID | $-\mu x_i + \beta \sum_j A_{ij}(1 - x_i)x_j$ |
| BIO | $\alpha - \delta x_i - \kappa \sum_j A_{ij}x_ix_j$ |
| POP | $-rx_i^b + \sigma \sum_j A_{ij}x_j^a$ |

Table 11: Parameters to reproduce the creation of the synthetic datasets.

| Dynamics | Initial condition | $T_{\text{Start}}$ | $T_{\text{End}}$ |
|---|---|---|---|
| KUR | Uniform $[0, 2\pi]$ | 0 | 1 |
| EPID | Uniform $[0, 1]$ | 0 | 2 |
| BIO | Uniform $[0, 1]$ | 0 | 1 |
| POP | Uniform $[-1, 1]$ | 0 | 10 |

For the KUR dynamics, oscillator phases are initialized uniformly in $[0, 2\pi]$ to cover the full angular domain. For EPID and BIO dynamics, node states are chosen in $[0, 1]$ to represent normalized concentrations or infection probabilities. For POP, instead, we chose to initialize nodes in the interval $[-1, 1]$ to better expose the effect of the polynomial term $x^a$, as including both positive and negative values yields richer trajectories. Regarding the temporal horizons $T_{\text{Start}}$ and $T_{\text{End}}$, we set them to allow each dynamics to display its full evolution.

The intermediate validation set used to tune the hyper-parameters of the SR algorithms is obtained by simulating the dynamics on an additional Barabási–Albert graph with 100 nodes and the same attachment parameter used for the training graphs. The graphs used to generate the three OOD test sets are a BA graph with 70 nodes (analogous to the one used for training but initialized with a different random seed), a Watts–Strogatz small-world graph with 50 nodes, and an Erdős–Rényi random graph with 100 nodes and an edge probability of 0.05. For validation and test datasets, we simulate the dynamics for $T = 1000$ steps.

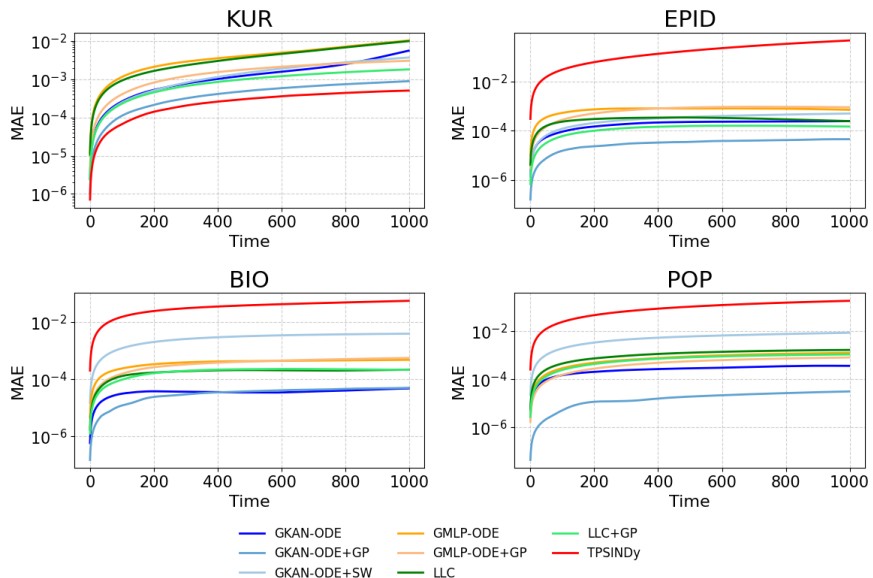

Figure 9: Evolution of test $\text{MAE}_{\text{traj}}$ over time for each assessed model on synthetic dynamics. Values are averaged over the three test sets.

All data-generating code, along with its random seeds, is provided in the codebase.

## B.2 Real-World Dataset and Preprocessing

The considered empirical datasets are based on the epidemiological spread of infectious diseases (SARS, COVID, H1N1), modeled by the worldwide airline network of human mobility between different countries, where each entry of the weighted adjacency matrix represents the traffic volume across regions. Refer to Gao & Yan (2022) for additional details. Regarding the COVID dataset, we normalize the values to the range $(-1, 1)$ using a `MinMax` scaler. The scaler is fitted only on the training set (the first 80% of the data) to prevent data leakage. We perform the same pre-processing steps for the H1N1 and SARS datasets before fine-tuning the coefficients of the learned symbolic formulas using neural models. During the evaluation phase, the same scaler is applied to transform the predicted values back to the original scale.

## C Additional Results and Ablation Studies

### C.1 Detailed Performance Analysis (MAE Time Evolution)

Figure 9 shows the test $\text{MAE}_{\text{traj}}$ over time obtained by the assessed models, including both the trained neural-based architectures and the distilled symbolic expressions. We can observe that, on EPID, BIO, and POP dynamics, GKAN-based models maintain the lowest error consistently over time, while for KUR, TPSINDy achieves the best performance.

### C.2 Additional Trajectory Metrics

To verify that our conclusions are not an artifact of the chosen error measure, we report three further trajectory metrics on the four synthetic systems: mean squared error ($\text{MSE}_{traj}$), root mean squared error ($\text{RMSE}_{traj}$), and mean absolute percentage error ($\text{MAPE}_{traj}$), computed analogously to $\text{MAE}_{traj}$ (Equation 14). Results, averaged over the three OOD test graphs (mean $\pm$ std), are reported in Tables 12–14. Across all three metrics the ranking is consistent with the $\text{MAE}_{traj}$ analysis: GKAN-ODE+GP attains the lowest error on every system and neural-based models dominate TPSINDy, which exhibits catastrophic error accumulation on EPID, BIO, and POP.

Table 12: Trajectory Mean Squared Error ($\text{MSE}_{traj}$), reported as mean ± standard deviation across synthetic test datasets.

| Model | KUR | EPID | BIO | POP |
|---|---|---|---|---|
| GKAN-ODE | $3.31 \times 10^{-5} \pm 4.22 \times 10^{-5}$ | $5.38 \times 10^{-8} \pm 9.10 \times 10^{-9}$ | $2.61 \times 10^{-9} \pm 8.53 \times 10^{-10}$ | $1.30 \times 10^{-7} \pm 3.76 \times 10^{-8}$ |
| GKAN-ODE+GP | $4.43 \times 10^{-7} \pm 4.41 \times 10^{-8}$ | $1.75 \times 10^{-9} \pm 5.33 \times 10^{-10}$ | $1.76 \times 10^{-9} \pm 2.79 \times 10^{-10}$ | $5.71 \times 10^{-10} \pm 6.51 \times 10^{-11}$ |
| GKAN-ODE+SW | $6.26 \times 10^{-6} \pm 2.02 \times 10^{-6}$ | $1.42 \times 10^{-7} \pm 5.34 \times 10^{-9}$ | $1.04 \times 10^{-5} \pm 1.77 \times 10^{-6}$ | $6.18 \times 10^{-5} \pm 2.24 \times 10^{-5}$ |
| GMLP-ODE | $6.85 \times 10^{-5} \pm 2.90 \times 10^{-5}$ | $1.15 \times 10^{-6} \pm 5.51 \times 10^{-7}$ | $2.60 \times 10^{-7} \pm 2.89 \times 10^{-8}$ | $1.54 \times 10^{-6} \pm 5.96 \times 10^{-7}$ |
| GMLP-ODE+GP | $6.10 \times 10^{-6} \pm 7.91 \times 10^{-7}$ | $7.51 \times 10^{-7} \pm 5.99 \times 10^{-8}$ | $2.17 \times 10^{-7} \pm 2.18 \times 10^{-8}$ | $5.21 \times 10^{-7} \pm 1.20 \times 10^{-7}$ |
| LLC | $6.83 \times 10^{-5} \pm 8.45 \times 10^{-6}$ | $2.09 \times 10^{-7} \pm 9.61 \times 10^{-8}$ | $6.31 \times 10^{-8} \pm 4.92 \times 10^{-9}$ | $3.93 \times 10^{-6} \pm 1.81 \times 10^{-6}$ |
| LLC+GP | $2.00 \times 10^{-6} \pm 1.54 \times 10^{-7}$ | $2.59 \times 10^{-8} \pm 2.97 \times 10^{-9}$ | $5.56 \times 10^{-8} \pm 1.83 \times 10^{-9}$ | $9.34 \times 10^{-7} \pm 1.68 \times 10^{-7}$ |
| TPSINDy | $1.73 \times 10^{-7} \pm 2.41 \times 10^{-8}$ | $7.50 \times 10^{-2} \pm 2.07 \times 10^{-2}$ | $1.99 \times 10^{-3} \pm 1.88 \times 10^{-4}$ | $1.70 \times 10^{-2} \pm 9.24 \times 10^{-4}$ |

Table 13: Trajectory Root Mean Squared Error ($\text{RMSE}_{traj}$), reported as mean ± standard deviation across synthetic test datasets.

| Model | KUR | EPID | BIO | POP |
|---|---|---|---|---|
| GKAN-ODE | $4.41 \times 10^{-3} \pm 3.69 \times 10^{-3}$ | $2.31 \times 10^{-4} \pm 1.94 \times 10^{-5}$ | $5.04 \times 10^{-5} \pm 8.43 \times 10^{-6}$ | $3.56 \times 10^{-4} \pm 5.33 \times 10^{-5}$ |
| GKAN-ODE+GP | $6.65 \times 10^{-4} \pm 3.33 \times 10^{-5}$ | $4.14 \times 10^{-5} \pm 6.28 \times 10^{-6}$ | $4.18 \times 10^{-5} \pm 3.39 \times 10^{-6}$ | $2.39 \times 10^{-5} \pm 1.34 \times 10^{-6}$ |
| GKAN-ODE+SW | $2.47 \times 10^{-3} \pm 3.86 \times 10^{-4}$ | $3.77 \times 10^{-4} \pm 7.06 \times 10^{-6}$ | $3.21 \times 10^{-3} \pm 2.76 \times 10^{-4}$ | $7.73 \times 10^{-3} \pm 1.41 \times 10^{-3}$ |
| GMLP-ODE | $8.04 \times 10^{-3} \pm 1.97 \times 10^{-3}$ | $1.04 \times 10^{-3} \pm 2.54 \times 10^{-4}$ | $5.09 \times 10^{-4} \pm 2.90 \times 10^{-5}$ | $1.21 \times 10^{-3} \pm 2.58 \times 10^{-4}$ |
| GMLP-ODE+GP | $2.46 \times 10^{-3} \pm 1.64 \times 10^{-4}$ | $8.66 \times 10^{-4} \pm 3.44 \times 10^{-5}$ | $4.65 \times 10^{-4} \pm 2.37 \times 10^{-5}$ | $7.17 \times 10^{-4} \pm 8.69 \times 10^{-5}$ |
| LLC | $8.25 \times 10^{-3} \pm 5.02 \times 10^{-4}$ | $4.46 \times 10^{-4} \pm 9.99 \times 10^{-5}$ | $2.51 \times 10^{-4} \pm 9.72 \times 10^{-6}$ | $1.93 \times 10^{-3} \pm 4.51 \times 10^{-4}$ |
| LLC+GP | $1.41 \times 10^{-3} \pm 5.39 \times 10^{-5}$ | $1.61 \times 10^{-4} \pm 9.13 \times 10^{-6}$ | $2.36 \times 10^{-4} \pm 3.86 \times 10^{-6}$ | $9.62 \times 10^{-4} \pm 8.92 \times 10^{-5}$ |
| TPSINDy | $4.15 \times 10^{-4} \pm 2.97 \times 10^{-5}$ | $2.71 \times 10^{-1} \pm 3.79 \times 10^{-2}$ | $4.45 \times 10^{-2} \pm 2.12 \times 10^{-3}$ | $1.30 \times 10^{-1} \pm 3.53 \times 10^{-3}$ |

## C.3 Ablation Study: Impact of Multiplicative Nodes in GKAN-ODE

In Figure 10, we show the performance obtained by GKAN-ODE models trained without multiplicative nodes (GKAN-ODE (no mult)) on the synthetic datasets. This architecture is equivalent to the original additive KAN model Liu et al. (2025b), depicted in Figure 2c. Since both architectures, with and without multiplicative nodes, can represent the same functions, the predictive performance of the respective raw models and the formulas extracted through GP are comparable across all the synthetic dataset. The decisive advantage of multiplicative nodes emerges during SW symbolic regression. In fact, as stated in Section 3.3, a KAN is an additive model by default, and for learning multiplicative interactions such as $x_1 x_2$ it requires learning logarithmic compositions ($e^{\ln x_1 + \ln x_2}$), or binomial expansion ($[(x_1 + x_2)^2 - (x_1 - x_2)^2]/4$). When we apply the SW algorithm, we attempt to replace each individual trained activation with a symbolic function from a candidate library. If the network has encoded $x_1 x_2$ via logarithmic compositions (Figure 2d), SW must recover $\ln(x_1)$, $\ln(x_2)$, and $e^{(\cdot)}$ across multiple activations and compose them correctly. With multiplicative nodes, the splines to be fit are all simple identities (Figure 2b), and the product structure is already encoded in the architecture. Figure 10 confirms this: the SW symbolic regression algorithm fails to identify the multiplicative term in EPID and BIO dynamics when multiplicative nodes are absent, but succeeds when they are present. On the contrary, the GP-based symbolic distillation method can easily discover this multiplicative term regardless of the underlying architecture, since it is model-agnostic.

## C.4 Ablation Study: Comparison with Original KAN Symbolic Regression

We compare the proposed algorithm for the Spline-Wise symbolic fitting with the original one introduced by the authors of KANs. Table 15 shows the complexity and $\text{MAE}_{traj}$ of the best-validated symbolic expressions inferred by the Original SW (OSW) method applied to the trained GKAN-ODE models, named GKAN-ODE+OSW. The results show that such a method is able to extract formulas that achieve very low $\text{MAE}_{traj}$, especially on EPID and BIO dynamics, but with very high structural complexity (thus making them less interpretable). Specifically, on EPID and POP our SW algorithm recovers expressions of complexity 10 and 16 with trajectory errors in the same order of magnitude as the ones obtained by OSW formulas, i.e., a factor-of-three reduction in complexity at no meaningful accuracy cost. The BIO dynamics illustrate a more nuanced case. The formula extracted through the OSW algorithm achieves a low trajectory error ($\text{MAE}_{traj} \approx 5.96 \times$

Table 14: Trajectory Mean Absolute Percentage Error ($\text{MAPE}_{traj}$), reported as mean $\pm$ standard deviation across synthetic test datasets.

| Model | KUR | EPID | BIO | POP |
|---|---|---|---|---|
| GKAN-ODE | $4.12 \times 10^{-4} \pm 1.75 \times 10^{-4}$ | $2.97 \times 10^{-4} \pm 2.53 \times 10^{-5}$ | $8.51 \times 10^{-5} \pm 7.91 \times 10^{-6}$ | $6.56 \times 10^{-3} \pm 3.87 \times 10^{-3}$ |
| GKAN-ODE+GP | $1.43 \times 10^{-4} \pm 1.67 \times 10^{-5}$ | $5.36 \times 10^{-5} \pm 9.43 \times 10^{-6}$ | $7.52 \times 10^{-5} \pm 3.05 \times 10^{-6}$ | $1.96 \times 10^{-4} \pm 6.95 \times 10^{-5}$ |
| GKAN-ODE+SW | $5.37 \times 10^{-4} \pm 6.68 \times 10^{-5}$ | $4.86 \times 10^{-4} \pm 4.83 \times 10^{-6}$ | $5.62 \times 10^{-3} \pm 5.00 \times 10^{-4}$ | $5.29 \times 10^{-2} \pm 1.27 \times 10^{-2}$ |
| GMLP-ODE | $1.86 \times 10^{-3} \pm 2.36 \times 10^{-4}$ | $1.34 \times 10^{-3} \pm 2.94 \times 10^{-4}$ | $7.68 \times 10^{-4} \pm 2.58 \times 10^{-6}$ | $1.37 \times 10^{-2} \pm 2.83 \times 10^{-3}$ |
| GMLP-ODE+GP | $5.09 \times 10^{-4} \pm 8.48 \times 10^{-5}$ | $1.23 \times 10^{-3} \pm 1.05 \times 10^{-4}$ | $6.62 \times 10^{-4} \pm 5.65 \times 10^{-5}$ | $1.02 \times 10^{-2} \pm 4.57 \times 10^{-3}$ |
| LLC | $1.73 \times 10^{-3} \pm 5.79 \times 10^{-4}$ | $5.80 \times 10^{-4} \pm 1.34 \times 10^{-4}$ | $3.98 \times 10^{-4} \pm 2.51 \times 10^{-5}$ | $1.89 \times 10^{-2} \pm 5.31 \times 10^{-3}$ |
| LLC+GP | $2.80 \times 10^{-4} \pm 1.40 \times 10^{-5}$ | $2.13 \times 10^{-4} \pm 1.86 \times 10^{-5}$ | $3.72 \times 10^{-4} \pm 8.03 \times 10^{-6}$ | $1.36 \times 10^{-2} \pm 5.98 \times 10^{-3}$ |
| TPSINDy | $8.53 \times 10^{-5} \pm 1.44 \times 10^{-5}$ | $2.59 \times 10^{-1} \pm 2.61 \times 10^{-2}$ | $7.75 \times 10^{-2} \pm 1.63 \times 10^{-3}$ | $9.30 \times 10^{-1} \pm 1.84 \times 10^{-1}$ |

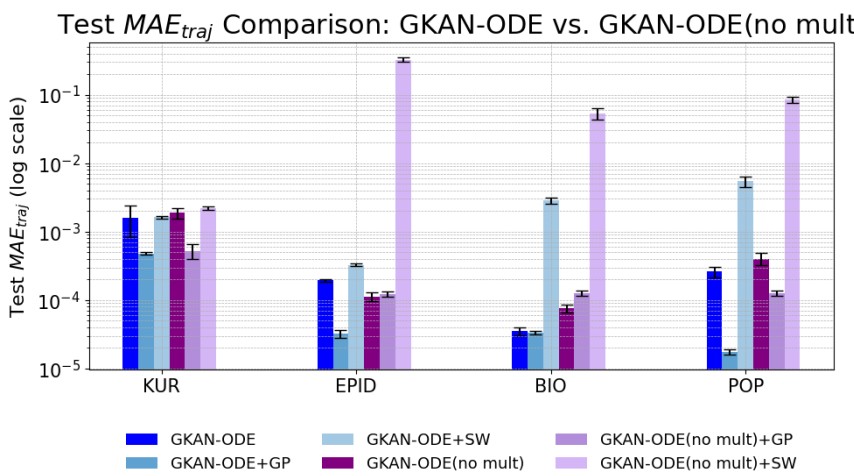

Figure 10: Performance comparison between GKAN-ODE models with and without multiplicative nodes.

$10^{-5}$) but at complexity 81. Our SW algorithm instead recovers $-0.5000 \cdot x_i + 1.0001 + \sum_j A_{ij}(-0.4899 \cdot x_i x_j)$ at complexity 6, which is structurally identical to the ground truth $1 - \frac{1}{2}x_i - \frac{1}{2}\sum_j A_{ij}x_i x_j$ up to a $\approx 2\%$ coefficient deviation. That small discrepancy accumulates over autoregressive integration, leading to a higher $\text{MAE}_{traj}$ ($\approx 2.84 \cdot 10^{-3}$) compared to OSW. However, identifying the right symbolic form with a slightly imprecise coefficient is scientifically more informative than a numerically accurate expression of complexity 81 that cannot be interpreted. The hyperparameter grid employed for validating the OSW algorithm is shown in Table 16.

## C.5 Topology Agnostic Neural Model

To demonstrate the importance of defining a topology-aware model that follows the structure of Equation 5, we additionally assess the performance of a topology-agnostic model, denoted as MLP-ODE. The MLP-ODE is a simple baseline that models node dynamics $\dot{\mathbf{x}}_i$ relying solely on the local state $\mathbf{x}_i$, effectively ignoring neighbor interactions and quantifying the specific contribution of topological information to the discovery process. Figure 11 illustrates the performance of the MLP-ODE model on the synthetic dynamics. The topology-agnostic baseline suffers from catastrophic error accumulation across most dynamics, demonstrating that the governing laws are inextricably linked to the specific network topology and cannot be resolved by simple curve-fitting or mean-field approximations. Notably, on the EPID and POP datasets, the naive MLP-ODE yields lower rollout errors than TPSINDy. This counterintuitive result highlights a critical limitation of restricted sparse regression: while TPSINDy is not compositional and fails by identifying incorrect interaction terms that lead to diverging trajectories, the MLP-ODE learns an approximated function that remains numerically stable. Table 17 shows the symbolic expressions extracted from the MLP-ODE models.

Table 15: Test $\text{MAE}_{\text{traj}}$ and structural complexity of the best-validated symbolic formulas extracted from the GKAN-ODE model. GKAN-ODE+OSW refers to the formulas obtained with the Original Spline-Wise algorithm, while GKAN-ODE+SW and GKAN-ODE+GP refer to the proposed Spline-Wise and Genetic Programming approaches. Values are averaged on three test graphs and the standard deviation is reported.

| Model | Dataset | Complexity | $\text{MAE}_{\text{traj}}$ |
|---|---|---|---|
| GKAN-ODE+OSW | KUR | 8 | $1.43 \cdot 10^{-3} \pm 2.39 \cdot 10^{-4}$ |
| | EPID | 49 | $1.40 \cdot 10^{-4} \pm 1.53 \cdot 10^{-5}$ |
| | BIO | 81 | $5.96 \cdot 10^{-5} \pm 4.82 \cdot 10^{-6}$ |
| | POP | 24 | $1.56 \cdot 10^{-2} \pm 8.28 \cdot 10^{-3}$ |
| GKAN-ODE+SW | KUR | 8 | $1.63 \cdot 10^{-3} \pm 7.67 \cdot 10^{-5}$ |
| | EPID | 10 | $3.27 \cdot 10^{-4} \pm 1.27 \cdot 10^{-5}$ |
| | BIO | 6 | $2.84 \cdot 10^{-3} \pm 3.17 \cdot 10^{-4}$ |
| | POP | 16 | $5.41 \cdot 10^{-3} \pm 1.00 \cdot 10^{-3}$ |
| GKAN-ODE+GP | KUR | 5 | $4.81 \cdot 10^{-4} \pm 2.46 \cdot 10^{-5}$ |
| | EPID | 6 | $3.22 \cdot 10^{-5} \pm 4.69 \cdot 10^{-6}$ |
| | BIO | 6 | $3.35 \cdot 10^{-5} \pm 1.91 \cdot 10^{-6}$ |
| | POP | 5 | $1.75 \cdot 10^{-5} \pm 1.44 \cdot 10^{-6}$ |

Table 16: Hyperparameter grid of the original Spline-Wise fitting algorithm.

| Hyperparameter | Values |
|---|---|
| Spline pruning threshold $\rho$ | $\{0.01, 0.05, 0.1\}$ |
| Grid range | $\{(-10, 10), (-5, 5)\}$ |
| Weight simple | $\{10^{-5}, 0.3, 0.7, 0.9\}$ |
| Symbolic library $\mathcal{O}$ | $\big[$`identity, square, cube, exp, abs, sin, cos, tan, tanh, ln, zero` $\big]$ |

## C.6 Robustness Analysis: Denoising algorithm

We adopted a more systematic anti-noise mechanism following the methodology proposed by Rudy et al. (2017). Specifically, rather than computing derivatives directly on the noisy observations, we perform a local polynomial interpolation of order $P = 3$ on the node states $\mathbf{x}(t)$. The time derivatives $\dot{\mathbf{x}}(t)$ are then computed from these smoothed polynomial proxies. This approach acts as a low-pass filter, preserving the underlying dynamics while suppressing the noise that typically destabilizes equation discovery algorithms. We evaluated this mechanism on the BIO dataset under the same signal-to-noise ratio conditions used in the main analysis. The results, reported in Table 18, demonstrate that this preprocessing step effectively stabilizes the performance of neural-based architectures. All neural models combined with symbolic regression (GP or SW) maintain trajectory errors in the order of $10^{-3}$ even at high noise levels (20 dB). In contrast, the baseline TPSINDy fails to recover accurate dynamics, exhibiting errors an order of magnitude higher

Table 17: Symbolic expression extracted from MLP-ODE trained models through GP on synthetic datasets.

| Dataset | Learned Expression | Complexity |
|---|---|---|
| KUR | $\exp(\tanh(\sin(\log(x_i))))$ | 4 |
| EPID | $1.2449 * (0.8962 * \tan(x_i) - 1.0000)^2$ | 5 |
| BIO | $|x_i^2 - \sin(x_i)|$ | 4 |
| POP | $-0.4389 * \log(\tanh(x_i) + 1.0000)$ | 5 |

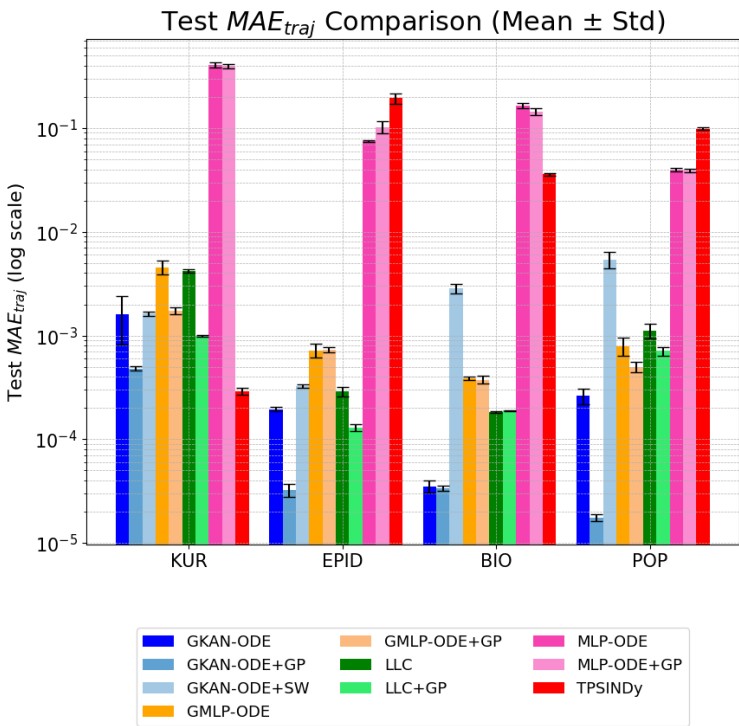

Figure 11: Performance comparison of MLP-ODE model against the other graph-aware models on synthetic datasets.

$(10^{-2})$, further highlighting the superior robustness of the proposed neural-symbolic pipeline in processing noisy data.

Table 18: Test $MAE_{\mathrm{traj}}$ (Mean $\pm$ Std) on the BIO dataset with noisy inputs, utilizing 3rd-order polynomial interpolation for robust derivative estimation.

| Model | 70 dB | 50 dB | 20 dB |
|---|---|---|---|
| GKAN-ODE+GP | $3.62 \times 10^{-3} \pm 2.25 \times 10^{-4}$ | $1.24 \times 10^{-3} \pm 1.55 \times 10^{-5}$ | $3.45 \times 10^{-3} \pm 3.11 \times 10^{-4}$ |
| GKAN-ODE+SW | $1.56 \times 10^{-3} \pm 2.22 \times 10^{-4}$ | $2.59 \times 10^{-2} \pm 2.89 \times 10^{-3}$ | $2.18 \times 10^{-3} \pm 3.21 \times 10^{-4}$ |
| GMLP-ODE+GP | $1.45 \times 10^{-3} \pm 1.59 \times 10^{-4}$ | $1.56 \times 10^{-3} \pm 1.10 \times 10^{-4}$ | $1.94 \times 10^{-3} \pm 1.40 \times 10^{-4}$ |
| LLC+GP | $1.34 \times 10^{-3} \pm 2.28 \times 10^{-4}$ | $1.66 \times 10^{-3} \pm 2.60 \times 10^{-4}$ | $2.74 \times 10^{-3} \pm 1.91 \times 10^{-4}$ |
| TPSINDy | $8.13 \times 10^{-2} \pm 1.97 \times 10^{-2}$ | $8.20 \times 10^{-2} \pm 5.02 \times 10^{-3}$ | $8.90 \times 10^{-2} \pm 4.70 \times 10^{-3}$ |

## C.7 Robustness Analysis: Derivative Estimation Method

To ensure that the superior performance of GKAN-ODE models is not an artifact of the specific numerical differentiation technique employed (i.e., the five-point stencil method), we conducted an ablation study using the Central Finite Difference method. We focused this analysis on all the synthetic datasets to evaluate model sensitivity to the quality of the target derivatives $\dot{\mathbf{X}}(t)$. The results, presented in Figure 12, demonstrate the robustness of GKAN-ODE method. The relative ranking of the models remains consistent with the main experimental results. Specifically, the GKAN-ODE approach (both the neural model and the GP distilled symbolic forms) continues to achieve the lowest trajectory error, consistently outperforming GMLP, LLC, and TPSINDy baselines. However, the SW symbolic models derived from GKAN-ODE achieve higher errors compared to the other symbolic equations, especially on POP dynamics. Finally, TPSINDy continues to exhibit significantly higher errors, confirming its struggle with long-term stability in this setting.

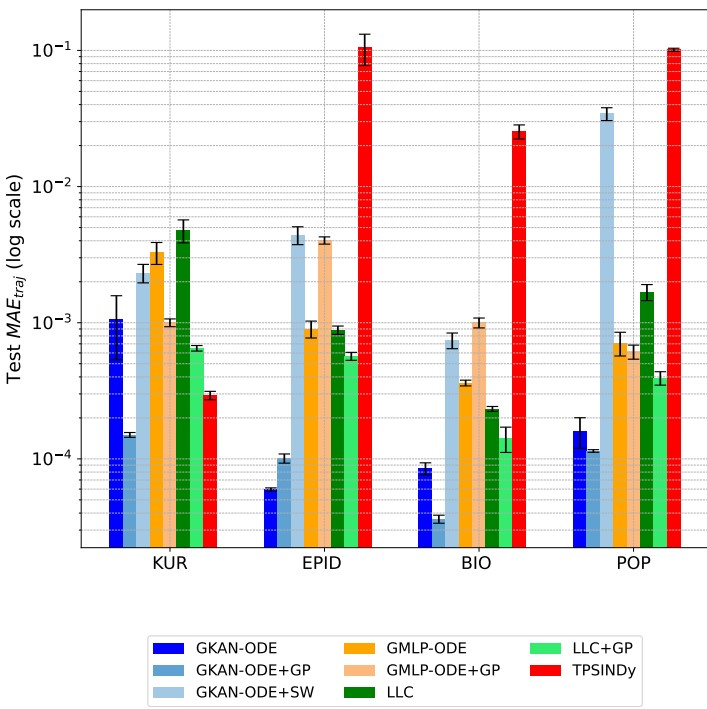

Figure 12: Performance comparison across datasets using the Central Finite Difference method for derivative estimation.

## C.8  Training and Validation Losses over the Epochs

Figure 13 shows training and validation losses of neural-based models (Equation 6) over the epochs on the synthetic dynamical systems. All the models show a similar pattern, that is, a steep drop in the early epochs followed by a plateau near zero, without any sign of overfitting. Notably, the GKAN-ODE model is the most stable, since its validation curve is the smoothest and flattest after convergence across all four systems. LLC is instead the least stable. Although it often reaches low final loss values, its validation loss oscillates violently and persistently right to the end of training in every system.

# D  Supplementary Information for Real-World Epidemic Dynamics

## D.1  Protocol for Country-Specific Coefficient Fine-Tuning

To account for the heterogeneity of real-world epidemic dynamics, we fine-tune the coefficients of the generic symbolic structures discovered by neural-based models (detailed in Table 4) for each node. Specifically, we replace scalar constant terms in the symbolic equations with trainable parameters and optimize them via gradient descent. The optimization is performed by retraining the expressions on each node's data using the first 80% of observations, with the subsequent 10% for validation, and leaving the final 10% for testing. Note that the LLC equation (and subsequent fine-tuning) is re-derived from scratch, as the original work does not report all the necessary coefficients needed for reproduction. Instead, the TPSINDy formula is the one provided in the original paper. However, for a fair comparison with neural-based equations, we re-executed the fine-tuning algorithm used in the TPSINDy paper only on the first 90% of observations. This leads to a set of coefficients very similar to the original one, but which does not depend on the entire dataset, which is crucial when evaluating the generalization capabilities of an ML model.

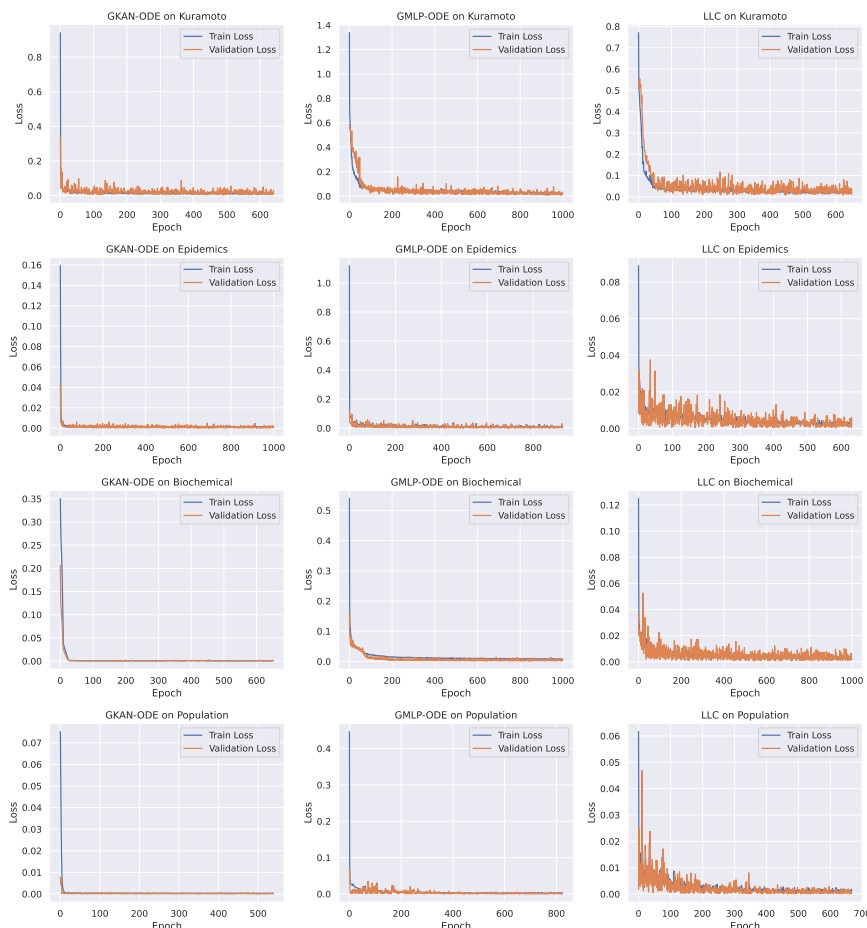

Figure 13: Training and validation losses over the epochs obtained by neural-based models on synthetic datasets. Note that the number of epochs varies depending on the model and dataset since early stopping is performed during training.

## E    Declaration on Generative AI

The author(s) have employed Generative AI tools for proofreading and improving the readability of figures and tables. No LLM was involved in the research design, experiments, analysis, or in the generation of any scientific content.

