# OpenReview forum: "Discovering Generalizable Governing Equations for Graph Dynamical Systems with Interpretable Neural Networks"
_TMLR — Accepted by TMLR_

### Review · Reviewer_cYSS · 2026-04-20

**Summary Of Contributions:**

Equation discovery using symbolic methods remains a challenging problem for graph-based dynamical systems. This occurs due to the dependence of structure on topology and a lack of verifiable data-dependent benchmarks. Furthermore, methods remain in nascent stages and a clear emphasis on interpretability is required. The paper aims to bridge this gap by proposing Graph Kolmogorov-Arnold Network-ODE (GKAN-ODE). GKAN-ODE adapts the KAN neural nework architecture of additive univariate spline-based activations towards graph-based dynamical systems. GKAN-ODE implements multiplicative nodes wherein half of the nodes in a KAN layer are designated with multiplicative operations. In addition, the work combines and utilizes GKAN-ODE in a comprehensive evaluation pipeline for equation discovery using both black-box and white-box methods. Dynamics are first extracted using a neural network for self and interaction dynamics prediction. The MLP then undergoes black/white-box regression to predict the equation form. In the case of black-box predictions, GP is utilized. For white-box interpretability analysis, the paper utiilizes SWR which conducts function fitting, optimal fit selection, accuracy-complexity balance using first-order change and model reconstruction of the best selected function. Evaluation is carried out using the MAE metric on an SR validation set and then OOD test set of unseen dynamics over the fixed topology. Experiments are carried out on a range of standard as well as synthetic datasets comparing GKAN-ODE to baselines.

**Audience:**

Yes

**Audience Explanation:**

The paper studies the important problem of equations discovery from the viewpoint of a graph-based dynamics perspective. GKAN-ODE focusses on an important application and motivates design decisions which may be further utilized in other domains of pure and applied sciences. The method, and its findings, may be of interest to the machine learning community, symbolic regression community as well as the ODE science/engineering community.

**Claims And Evidence:**

No

**Claims Explanation:**

* **Spline-Wise Regression:** An area of concern remains the SWR. It remains unclear what purpose it serves and how is it more interpretable. Black-box GP is found to be performant but SWR does not provide a comprehensive outlook on interpretability. At its core, the work utilizes SWR to interpet the high-level equation mismatch. A faithful comparison of SWR with alternative methods remains absent. Can SWR be compared to similar spline-based regression methods such as cubic splines, penalized splines, etc? Additionally, SWR could also be evaluated with learnable regression methods such as VAEs or gradient-boosted regression. Authors claim in Section 5.2 that SWR, compared to OSW, provides a better accuracy-complexity tradeoff. However, this does not remain well-justified. OSW is performant in accurate predictions in Table 12. Compared to SWR, for a ~10X drop in complexity, OSW still retains its strong prediction accuracy. Could the authors clarify how SWR provides a better trade-off over OSW?

* **Multiplicative Nodes:** It remains unclear as to what is the role and utility of multiplicative nodes in GKAN-ODE. From what I understand, multiplicative nodes add more expressivity and are analogous components to bridge the gap with expressive MLPs. But what is their core efficacy? How is the partitioning of multiplicative nodes different from MLPs in LLC? Figure 9 evaluates the utility of multiplicative nodes but there does not seem to a be consistent answer. For GKAN-ODE, addition or removal of mutliplicative nodes makes little to no difference. However, these are found to be beneficial for GP and SWR. Could the authors explain the role of multiplicative nodes and their overall contribution to the evaluation pipeline.

* **Empirical Evaluation:** Experiments and evluation presented in the paper can be made more comprehensive. SWR, being the central technical contribution of GKAN, should be thoroughly compared to pre-existing spline-wise regression methods or gradient-based methods. In addition to MAE$_{\text{traj}}$, authors should consider additional metrics for a comprehensive study that would make the evaluation more thorough. These could include correlation, MSE and percentage deviation around mean.

**Requested Changes:**

* Comparison of SWR with pre-existing spline-based regression methods and discussion on the role and utility of SWR. Clarification and comparison of SWR with OSW for accuracy and complexity.

* Empirical evaluation consisting of additional metrics such as MSE, correlation or percentage deviation around mean.

* Explanation and discussion  on the role of multiplicative nodes. Authors should explain the motivation behind addition of these nodes, their overall beneficial aspects and whether they boost or hinder expressivity.

---

> ### Author Response · Authors · 2026-06-04
> **Answer Reviewer cYSS (part 1)**
>
> ## RC1 - **Purpose and interpretability of SW Symbolic Regression Algorithm.**
>
> The goal of the Spline-Wise (SW) *symbolic regression* algorithm is to convert a trained KAN into a human-readable mathematical expression by performing symbolic regression on each individual univariate activation function (parameterized as B-splines during training, as in the KAN original paper), and then composing the resulting symbolic pieces according to the KAN's architecture into a mathematical formula. To illustrate this, suppose we trained a KAN with two input nodes, one intermediate node and one output node (represented by the array [2, 1, 1]) to approximate the expression $y=(x_1+x_2)^2$. Once training is complete, the SW algorithm should identify the two activations entering the intermediate neuron as identity functions, so the neuron computes $\ell_1 = x_1 + x_2$. Then, the algorithm should fit the output activation as quadratic function, yielding the final formula $y = \ell_1^2$ as expected.
>
> In contrast, a GP-based SR method is *model-agnostic*: it receives a set of input-output pairs $(x_i,y)$ sampled from the trained network and searches for a compact symbolic expression matching those pairs, without any knowledge of the network's internal structure. While GP can be highly effective, it provides no window into how the model arrived at its answer. SW is therefore complementary to GP: it offers a transparent, node-by-node account of what the KAN has learned.
>
> **Why the comparison to cubic splines, VAEs, or gradient boosting is not applicable.** The methods mentioned (cubic splines, penalized splines, VAEs, gradient-boosted regressors) are *not* symbolic regression techniques and do not produce human-readable mathematical expressions. They would yield numerical approximations, not interpretable governing laws, so they are outside the scope of this comparison. The use of splines to parameterise the KAN activation functions is not a fundamental requirement of our algorithm: it is simply inherited from the original KAN formulation by Liu et al. (2025), which adopts B-splines as a convenient and flexible differentiable basis. Other parameterisations of the univariate activations, such as radial basis functions, wavelet functions, or Chebyshev polynomials, are equally compatible with the proposed procedure. **The core idea of the algorithm is to perform symbolic regression independently on each activation function, whatever its parameterisation, and then compose the resulting symbolic expressions according to the network's architecture.** To clarify this further, we added the following phrase in Section 3.3 when introducing the GKAN-ODE model: "For convenience, throughout this work we use the term *splines* to denote the generic KAN activation functions. [...]". This is intended to make clear that the words *splines* and *activations* refer to the same concept throughout the paper. Furthermore, we added in Section 2.3 pointers to relevant works that studied different parametrizations for KANs' activations.
>
> **Clarification on the SW vs. OSW trade-off.**
> We acknowledge that the phrasing of our trade-off claim deserves clarification. On EPID and POP, our SW algorithm recovers expressions of complexity 10 and 16 with trajectory errors in the same order of magnitude as the formulas obtained through the original OSW (complexity 49 and 24, Table 15 of the new manuscript (former Table 12)), i.e., a factor-of-three reduction in complexity at no meaningful accuracy cost. The BIO dynamics illustrate a more nuanced case. OSW achieves low trajectory error ($\\text{MAE}\_{traj} \\approx 5.96 \\times 10^{-5}$) but at complexity 81. Our SW algorithm instead recovers $-0.5000 \\cdot x_i + 1.0001 + \\sum_j A_{ij}(-0.4899 \\cdot x_i x_j)$ at complexity 6, which is structurally identical to the ground truth $1 - \\frac{1}{2}x_i - \\frac{1}{2}\\sum_j A_{ij} x_i x_j$ up to a ~2% coefficient deviation. That small discrepancy accumulates over autoregressive integration, leading to a higher $\\text{MAE}_{traj}$ ($\\approx 2.84 \\cdot 10^{-3}$) compared to OSW. However, identifying the right symbolic form with a slightly imprecise coefficient is scientifically more informative than a numerically accurate expression of complexity 81 that is hardly interpretable.
>
> We modified Section 3.4.2 to better explain the purpose of the proposed algorithm, the reason behind its development and its benefits compared to the OSW approach. Additionally, we expanded Appendix C.4 with a more detailed discussion on the comparison between the formulas extracted with our SW algorithm and the one obtained through OSW. Finally, we specified in Section 4.1 that the complexity of the retrieved formulas is computed using the `count_ops` method from the `sympy` library, which weights the number of mathematical operations in a given mathematical expression.

---

> ### Author Response · Authors · 2026-06-04
> **Answer Reviewer cYSS (part 2)**
>
> ## RC2 - Additional Metrics
>
> In response to this suggestion, we have computed additional metrics including trajectory Mean Squared Error ($\\text{MSE}\_{traj}$), Root Mean Squared Error ($\\text{RMSE}\_{traj}$), and Mean Absolute Percentage Error ($\\text{MAPE}\_{traj}$) for all models across the four synthetic dynamical systems. These are reported in Appendix C.2. The results are consistent with the conclusions drawn from $\\text{MAE}\_{traj}$ in the main paper.
>
> ## RC3 - Role of multiplicative nodes
>
> **Motivation** Because each neuron in a standard KAN is defined as a *sum* of univariate functions, learning a multiplicative interaction $y = x_1x_2$ requires the network to exploit indirect representations. For example, a [2, 1, 1] KAN (two input nodes, one intermediate node and one output node) can learn $x_1x_2$ via a logarithmic composition: the intermediate neuron computes $\\ell_1 = \\ln(x_1) + \\ln(x_2)$, and the output activation learns $y = e^{\\ell_1}$. Alternatively, the network may use a binomial expansion: $[(x_1+x_2)^2 - (x_1-x_2)^2]/4$. Both routes require overly-complex activations.
>
> Now consider the same [2, 1, 1] KAN, but with the intermediate neuron replaced by our proposed multiplicative node. Instead of learning $\\ln(x_1)$ and $\\ln(x_2)$, the two input splines only need to learn identity functions, and the multiplicative node computes $\\ell_1 = x_1x_2$ directly. Also the output neuron's activation then needs to learn a simple linear function, making the learning task substantially simpler.
>
> We added to Section 3.3 a Figure (Figure 2) showing the differences between a standard additive KAN and the proposed KAN with multiplicative nodes, also illustrating how the two architectures learn the multiplicative term $x_1x_2$ following the examples given above.
>
> Since both architectures, with and without multiplicative nodes, can represent the same functions, their *predictive performance* is comparable (as seen in Figure 10 for the raw models and GP-based formulas performance). **The decisive advantage of multiplicative nodes emerges during Spline-Wise symbolic regression**. When we apply the SW algorithm, we attempt to replace each individual trained activation with a symbolic function from a candidate library. If the network has encoded $x_1x_2$ via logarithmic compositions, SW must recover $\\ln(x_1)$, $\\ln(x_2)$, and $e^{(\\cdot)}$ across multiple activations and compose them correctly. With a multiplicative node, the three splines to be fit are all simple identities (or near-identities), and the product structure is already encoded in the architecture. Figure 10 confirms this: the SW symbolic regression algorithm fails to identify the multiplicative term in EPID and BIO dynamics when multiplicative nodes are absent, but succeeds when they are present. GP is unaffected because it treats the network as a black box and discovers the product directly from input-output data.
>
> This discussion on the role of multiplicative nodes in the context of SW symbolic regression has been additionally inserted in Appendix C.3.
>
> **Comparison with LLC's multiplicative structure.** LLC enforces a fixed and pre-defined interaction structure: $\\hat{G}(x_i, x_j) = \\text{MLP}\_1(x_i, x_j) + \\text{MLP}\_2(x_i) \\cdot \\text{MLP}\_3(x_j)$. This imposes a multiplicative bias on the interaction term regardless of whether the true dynamics are multiplicative, which may introduce unnecessary complexity for purely additive systems. By contrast, our approach embeds multiplicative nodes *within* the KAN layers alongside standard additive nodes. **Combined with the sparsity regularisation during training, the network automatically learns whether to activate or suppress the multiplicative nodes based on the data.** As we show in Appendix C.3, training with sparsity effectively prunes multiplicative nodes when the dynamics are purely additive and retains them when they are essential.
>
> ### References
>
> Liu, Ziming, et al. "KAN: Kolmogorov–arnold networks." *International conference on learning representations* (ICLR). 2025\.

---

### Review · Reviewer_jSu4 · 2026-04-26

**Summary Of Contributions:**

Summary: This work studies SysID for graph dynamical systems using Kolmogorov-Arnold Network and also studies symbolic regression approach for resulting NN models.  Further, using long-term model misspecification evaluation and out-of-distribution test, this work tests the proposed approach against other existing methods.

**Audience:**

Yes

**Audience Explanation:**

SysID for graph dynamical systems and the integration of SR are definitely important studies in ML.

**Claims And Evidence:**

No

**Claims Explanation:**

While the paper is generally clearly written and the topic is interesting enough, I have several concerns.

1. Mathematical statements in general are not rigorous enough (I don’t mean it needs more math, I mean the expressions are not rigorous enough). Starting from page 2 \mathcal{F}, f etc. are not clearly defined (also, the statement mentioning the difference between standard regression and SR is not clear or convincing); page 3 what is d_0 = |x|?  Absolute value?  Page 4, you may need comma for \forall t, A(t)=A.  Page 5 (5); is it only for one fixed N?  Or does it depend on data set?
Page 6, what is the need for subscript \phi for f_\phi?  Is (a,b,c,d) for \theta or for \theta^*_{f,\phi}?
(7) is unclear for me because of the aforementioned reason.

2. Although the work explains what their contributions are, it is a bit hard to see why we need this work from the context of existing studies.  The author may want to clarify existing issues on why symbolic dynamics is important for graph dynamical systems, why existing approach cannot solve it, what is the situation of current evaluation methodology; and state why KAN is best-suited for this purpose etc.  Some chart may be helpful.  Currently, while I can see the work contains several contributions, I cannot clearly see their connections and why these are required as a logical consequence.

3. While the experiments are interesting, I would like to see more analysis on why the proposed approach worked better.

I believe a bit careful revision of the paper will make this work very interesting to wider audience.

**Requested Changes:**

Please refer to my comments above and revise the paper accordingly.

---

> ### Author Response · Authors · 2026-06-04
> **Answer Reviewer jSu4 (part 1)**
>
> ## RC1 - Notation and mathematical rigor
>
> **Formal definition of symbolic regression (Section 2.1).** We have rewritten the opening of Section 2.1 to provide a more formal definition of symbolic regression. In particular, we now explicitly introduce the set of admissible primitive operators $\\mathcal{O}$ and define the search space $\\mathcal{F}_{\\mathcal{O}}$ as the class of closed-form expressions that can be constructed by finite composition of elements of $\\mathcal{O}$. Furthermore, we clarified the distinction between parametric regression and symbolic regression.
>
> **Minor notation fixes.**
> - We replaced the ambiguous $d_0=∣\\mathbf{x}∣$ wording on page 3 (now page 4) with $d_0 = \\dim(\\mathbf{x})$, i.e., the dimensionality of $\\mathbf{x}$.
>
> - We added the missing comma in the static-topology statement on page 4 (now page 5), which now reads $\\forall\ t, \ A(t)=A$.
>
> - We clarified Equation (6) (the training loss) by making explicit that $N$ and $T$ depend on the dataset.
>
> **Formal description of the Spline-Wise symbolic regression procedure.** We agree with the reviewer that the original formulation was unclear. We have rewritten Section 3.4.2 (Steps 1 and 2) to make the procedure mathematically explicit. In short, the affine-transformed candidate is now written as $\\tilde{f}(x; \theta) = a \\cdot f(b \\cdot x + c) + d$, with $\\theta = (a,b,c,d) \\in \\mathbb{R}^4$. The spurious subscript $\\phi$ on $f$ has been removed, since the spline-specific quantity is the parameter vector $\\theta^{\\star}\_{f,\\phi}$, not the candidate function itself. The subscript $\\phi$ is now present only when defining $f^{\\star}\_{\\phi, \\gamma}$, that is, the candidate expression selected at regularization level $\\gamma$ for spline $\\phi$. Note that the former Equation (7) is now Equation (10). Furthermore, we updated Algorithm 1 in Appendix A.4 to make it consistent with the new notation of Section 3.4.2.

---

> ### Author Response · Authors · 2026-06-04
> **Answer Reviewer jSu4 (part 2)**
>
> ## RC2 - Positioning of the contribution
>
> **Positioning of GKAN-ODE**. To make this positioning more concrete, we added a qualitative diagram to the revised paper (Figure 1) that places different methods for learning on graph dynamical systems in the expressivity–interpretability plane, mirroring the style of Koenig et al. (2024). The discussed models are Graph Neural ODE (GNODE), TPSINDy and Neural-based models (the proposed GKAN-ODE and LLC). In summary, GNODEs achieve high expressivity but minimal interpretability (opaque latent representations). TPSINDy is highly interpretable by construction, but limited in expressivity (restricted to its symbolic library). Neural-based model attempt to balance the two extremes by training a flexible neural model and then applying a post-hoc symbolic regression step to distill the functions learned by the model. GKAN-ODE moves further along the interpretability axis compared to LLC, since the KAN architecture exposes each univariate activation function individually, allowing direct inspection of the model's internal components and enabling the structure-aware SW symbolic regression, which is not meaningful with MLP-based architectures.
>
> **Why current evaluation methodology is inadequate.** Two critical aspects are largely overlooked in the existing literature. First, symbolic expressions are typically evaluated on trajectories from the *same* graph instance used during training. A model that overfits to a specific topology may achieve low error in-distribution while failing to generalise to any other networks. Second, the hyperparameters of the SR step are rarely validated in a principled way. Both of these gaps can mask poor scientific utility behind seemingly strong numerical results. Our evaluation pipeline addresses both issues by introducing an explicit out-of-distribution (OOD) test on unseen graph topologies and a systematic validation procedure for SR hyperparameters.
>
> We added to the introduction a short paragraph explaining these limitations in current works and how we propose to handle them with our proposed methodology.
>
> **Why KANs are well-suited for this domain.** KANs replace weights and fixed nonlinear activations of MLPs with *learned* univariate functions on edges (parameterised as B-splines) and sum aggregation on nodes. This architectural design is well-suited specifically for dynamical systems (as investigated by Koenig et al., 2024). In fact, physical governing equations are generally composed of smooth, structured nonlinear terms, such as trigonometric functions in oscillator models, exponentials in reaction kinetics, or power laws in population dynamics. A network whose elementary building blocks are themselves smooth, learnable nonlinear functions is therefore naturally aligned with this class of targets. On the contrary, an MLP must construct this non-linearity through the composition of many simple fixed activations across many layers, while a KAN can represent the same non-linearity compactly in few learned splines. Furthermore, once training is complete, each univariate activation can be inspected and matched to a symbolic function, providing a transparent, node-by-node view of the model's learned dynamics. This is an interpretable window *in addition to* the GP-based post-hoc SR. This description is now included in Section 2.3 when introducing KANs.
>
> ## RC3 - Additional Analyses
>
> We appreciate the reviewer's interest in our extensive experimental evaluation. To provide a more comprehensive study, we have computed additional metrics including trajectory Mean Squared Error ($\\text{MSE}\\_{traj}$), Root Mean Squared Error ($\\text{RMSE}\_{traj}$), and Mean Absolute Percentage Error ($\\text{MAPE}\_{traj}$) for all models across the four synthetic dynamical systems. These results, reported in Appendix C.2, are consistent with our previous findings. Furthermore, the Appendix includes deeper analyses on the temporal evolution of the MAE (C.1), the specific utility of multiplicative nodes (C.3) and our proposed SW symbolic regression algorithm (C.4) compared to the one proposed by KAN’s authors, as well as ablation studies on graph bias (C.5), performance under noisy conditions (C.6), and overfitting (C.8).
>
> We remain open to further suggestions for analyses that could strengthen the impact of this work.
>
> ### References
>
> Koenig, Benjamin C., Suyong Kim, and Sili Deng. "KAN-ODEs: Kolmogorov–Arnold network ordinary differential equations for learning dynamical systems and hidden physics." *Computer Methods in Applied Mechanics and Engineering* 432 (2024): 117397\.

---

> > ### Comment · Reviewer_jSu4 · 2026-06-25
> > **Thank you for the response**
> >
> > My concerns have been addressed.

---

### Review · Reviewer_FZxH · 2026-05-26

**Summary Of Contributions:**

This paper proposes a new Symbolic Regression method, called *GKAN-ODE*, to model temporal evolution of variables distributed on the nodes of a graph. The proposed method consists in:
1. modeling the evolution of the variable $x_i$ of each node $i$ as the sum of a "self" term $H(x_i)$ and a sum of "interaction" terms $A_{ij} G(x_i, x_j)$, for all the neighbors $j$ of the node $i$;
2. perform the symbolic training of the functions $G$ and $H$ through Kolmogorov-Arnold Networks (KANs);
3. possibly simplify the resulting functions $G$ and $H$ by performing "spline-wise symbolic regression" (SW).

Contrary to modeling $G$ and $H$ with neural networks (like MLPs), KANs networks are interpretable functions.

To evaluate their method, the authors present an experimental design centered on the evaluation of a model through out-of-distribution datasets. Finally, they evaluate it on synthetic datasets and a real-world dataset (epidemic data).

**Audience:**

Yes

**Audience Explanation:**

The proposed model is relevant for the Machine Learning community, and it is novel enough.

**Claims And Evidence:**

Yes

**Claims Explanation:**

# Choice of model

KANs are a valid choice in Symbolic Regression, compared to the usual neural networks (as MLPs), since they are directly interpretable as functions, while MLPs are usually regarded as block boxes.

# Theoretical ground

The Kolmogorov-Arnold representation theorem guarantees that any *continuous* function (not necessary smooth) can be represented in a specific way described in Equation (1), which is exactly how the KANs are designed. So, if the target function is continuous, then the task can be achieved (theoretically).

# Methodology

The proposed evaluation pipeline (see Figure 2) is convincing: the evaluation (validation and test sets) is performed on graphs that have not been observed during training.

**Requested Changes:**

# Size of the model

To which extent is the proposed method scalable? (large $d_{\mathrm{in}}$ or $d_{\mathrm{out}}$)
What is the training time?

# Set of splines

How did you choose the set of splines? What do you recommend to the reader?

# Inputs space

According to the usual statement of KA representation theorem, the inputs should belong to $[0, 1]^d$. Is this a limitation? How to deal with data that belong to $\mathbb{R}^d$ without any specific restriction?

# Novelty

Application to graphs: explain the novelty compared to KANs.

# Train/test discrepancy

Do you observe overfitting? Could you provide train/test losses at least on one example?

---

> ### Author Response · Authors · 2026-06-04
> **Answer Reviewer FZxH**
>
> ## RC1 - Size of the model
>
> The proposed GKAN-ODE (being a composition of KANs) inherits their same scaling behaviour, which is in turn derived by the Kolmogorov-Arnold Representation Theorem. The authors showed that KANs follow an exponential neural scaling law with exponent $\\alpha=-4$, whereas MLPs are more inefficient $\\alpha=-1$ (Liu et al., 2025, Section 2.3). That is, a ten-fold parameter increase gives theoretically a 10,000 error reduction, making KAN one of the most scalable neural architectures. We performed a comparison of the performances and their parameter count in Figure 4, and observed a compatible behavior of the better scalability of GKAN-ODE.
>
> Concerning the feature dimensions, the current focus was on univariate graph dynamical systems. This is an acknowledged limitation, and the method is applicable to any multivariate system and larger $d$.
>
> Unfortunately, we did not record the per-epoch training time of the tested models, but we only saved the timestamp of each Optuna trial. Moreover, because we implemented early stopping, a fully fair comparison of training times across models is not possible, as one model may halt training earlier than another depending on the patience parameter. Nevertheless, the longest time observed across all Optuna trials was 29 minutes, achieved by GKAN-ODE on POP dynamics, indicating that the training phase is not computationally heavy. All models were accelerated using an NVIDIA L4 GPU.
>
> It is also worth recalling that KANs are known to be slower than equivalent MLPs (Liu et al., 2025, Section 2), arguably for less-efficient implementations not optimized for parallel computing. For example, this has been partially addressed in efficient-kan ([https://github.com/Blealtan/efficient-kan](https://github.com/Blealtan/efficient-kan)), although this implementation makes the original KAN sparsity loss not applicable.
>
> ## RC2 - Set of splines
>
> We parametrize KANs' activations as B-splines with residual activation, inheriting the original formulation proposed by Liu et al., 2025. We added to Section 2.3 some pointers to relevant works investigating alternative parametrizations for KANs’ activations, including radial basis functions, Chebyshev polynomials, and wavelet functions.
>
> ## RC3 - Inputs space
>
> This question has been previously addressed in the KAN paper (Liu et al., 2025, Section 2.3). Briefly, this is handled by simple input normalization for bounded data, adaptive grid extension for unbounded data, and the fact that KAN deliberately uses smooth B-splines and deep architectures. This is to go beyond the classical KA theorem to make it computationally viable on any domain in $\\mathbb{R}^d$ (see also the definition of the KAN layer $\\Phi_l: \\mathbb{R}^{d_{in}} \\rightarrow \\mathbb{R}^{d_{out}}$ in Equations 2 and 3 in the revised manuscript).
>
> ## RC4 - Novelty
>
> KANs have already been applied for learning on dynamical systems (Koenig at al., 2024). We propose instead a **KAN-based model explicitly tailored for graph dynamical systems**, where the network topology shapes the behavior of the system. This required the development of a message-passing-based KAN model (Equation (5)), a training algorithm for learning the underlying ODE, and the post-hoc symbolic regression step. To the best of our knowledge, the use of KANs has been limited to other graph-based tasks (Bresson et al., 2025), not to the specific challenge of discovering underlying temporal dynamics. We added a paragraph in Section 2.3 explaining why KANs are well-suited for learning on (graph) dynamical systems.
>
> ## RC5 - Train/test discrepancy
>
> We added a new Appendix section (C.8), showing the plots of training and validation losses over the epochs obtained by neural-based models on the synthetic datasets. The results show that all the models show a similar pattern, that is, a steep drop in the early epochs followed by a plateau near zero, without any sign of overfitting. Notably, the GKAN-ODE model is the most stable, since its validation curve is the smoothest and flattest after convergence across all four systems.
>
> ### References
>
> Liu, Ziming, et al. "KAN: Kolmogorov–arnold networks." *International conference on learning representations* (ICLR), 2025\.
>
> Koenig, Benjamin C., Suyong Kim, and Sili Deng. "KAN-ODEs: Kolmogorov–Arnold network ordinary differential equations for learning dynamical systems and hidden physics." *Computer Methods in Applied Mechanics and Engineering* 432 (2024): 117397\.
>
> Bresson, Roman, et al. "KAGNNs: Kolmogorov-Arnold Networks meet Graph Learning." *Transactions on Machine Learning Research* (TMLR), 2025

---

> > ### Comment · Reviewer_FZxH · 2026-06-28
> >
> > I have carefully read the other reviews, which, I think, do not raise any major issue. I am also happy with the authors' response. I do not have any more concern about the paper in its current form. So, I maintain my review as such.

---

### Author Response · Authors · 2026-06-04
**Message to Action Editor and Reviewers**

## General Comment

Dear Action Editor and Reviewers,

We would like to sincerely thank you all for your investment of time, meticulous reading, and highly constructive feedback. Your insightful critiques have been valuable in helping us sharpen our claims and strengthen the overall quality of our submission.

In response to your comments, we have substantially updated our paper. We have uploaded a revised version of the manuscript, where all major modifications and additions are explicitly highlighted in blue (while removals are marked with a red strikethrough) for ease of tracking.

Below is a high-level summary of the core improvements made in this revision:

###

* **Formally defined the symbolic regression** search space (Sec 2.1) and clarified the step-by-step mathematical formulation of the Spline-Wise (SW) algorithm (Sec 3.4.2, Alg 1).
* Added **Figure 1** (mapping methods on an Expressivity–Interpretability plane) and **Figure 2** (illustrating how multiplicative nodes streamline complex compositions).
* **Expanded analyses** in the Appendix C with multiple trajectory metrics (MSE, RMSE, MAPE) and overfitting discussion.
* **Enhanced conceptual clarity and positioning,** deepened discussions regarding the physical alignment of KANs, the structural complexity trade-offs between SW and OSW, and how sparsity regularization dynamically prunes or retains multiplicative nodes.

We have posted detailed responses to every individual comment and requested change (RC) directly under each reviewer's respective thread.

We believe these revisions fully address your questions and have significantly elevated the clarity and empirical backing of our work. We remain entirely open to further suggestions and look forward to an engaging discussion\!

Best regards,
The Authors

---

### Decision · Action_Editor_YWVt · 2026-07-02

**Recommendation:** Accept as is

**Audience:**

Yes

**Audience Explanation:**

Yes. Symbolic equation discovery for graph dynamical systems, together with the proposed OOD generalization benchmark, is relevant to the TMLR readership.

**Claims And Evidence:**

Yes

**Claims Explanation:**

Following the rebuttal, the reviewers do not have any outstanding concerns on the central claims made by this paper. FZxH found the out-of-distribution evaluation pipeline convincing and the KAN choice well motivated. jSu4 and cYSS initially answered No, citing insufficient mathematical rigor, weak positioning, and missing analyses. Both confirmed post-revision that their concerns were resolved. The authors formalized the symbolic regression search space and the Spline-Wise algorithm, added the expressivity-interpretability positioning figure, reported additional trajectory metrics (MSE, RMSE, MAPE), and added an overfitting analysis. Some baseline comparisons requested by cYSS were not added. However, cYSS did not treat this as blocking and judged the methodology sound.